# Adaptive Moments are Surprisingly Effective for Plug-and-Play Diffusion Sampling

**Christian Belardi**    **Justin Lovelace**    **Kilian Q. Weinberger**    **Carla P. Gomes**
Cornell University
`ckb73@cornell.edu`

## Abstract

Guided diffusion sampling relies on approximating often intractable likelihood scores, which introduces significant noise into the sampling dynamics. We propose using adaptive moment estimation to stabilize these noisy likelihood scores during sampling. Despite its simplicity, our approach achieves state-of-the-art results on image restoration and class-conditional generation tasks, outperforming more complicated methods, which are often computationally more expensive. We provide empirical analysis of our method on both synthetic and real data, demonstrating that mitigating gradient noise through adaptive moments offers an effective way to improve alignment. Our code is available at `https://github.com/christianbelardi/adam-guidance`.

## 1 Introduction

Diffusion models (Sohl-Dickstein et al., 2015; Song & Ermon, 2019; Ho et al., 2020) have become one of the most prevalent approaches to generative modeling, achieving state-of-the-art results in many domains, including text-to-image synthesis (Rombach et al., 2022; Ramesh et al., 2022; Saharia et al., 2022a), image-to-image translation (Saharia et al., 2022b), audio generation (Kong et al., 2020; Liu et al., 2023), video synthesis (Ho et al., 2022; Brooks et al., 2024), and molecular design (Hoogeboom et al., 2022; Watson et al., 2023).

Plug-and-play conditional generation enables sampling from a conditional distribution $p(x|y)$ using a diffusion model trained only on the marginal distribution $p(x)$. These methods guide the sampling process toward desired conditions $y$ without task-specific training. While some approaches like Classifier Guidance (CG) (Dhariwal & Nichol, 2021) train time-aware models directly on the diffusion latents, many plug-and-play methods leverage existing models that operate on clean data – whether analytical forward operators for inverse problems or pre-trained classifiers – making them highly flexible for diverse applications.

The plug-and-play guidance literature has evolved from early methods like Diffusion Posterior Sampling (DPS) (Chung et al., 2023) to increasingly sophisticated techniques. Recent work such as Universal Guidance for Diffusion Models (UGD) (Bansal et al., 2023) and Training-Free Guidance (TFG) (Ye et al., 2024) compose multiple algorithmic components, gradient computations in both latent and data spaces, Monte Carlo approximations, and iterative refinement procedures.

At the heart of plug-and-play guidance lies the challenge of incorporating the desired condition into the diffusion sampling process. Sampling in diffusion models can be understood as annealed Langevin dynamics using the score function $\nabla_{x_t} \log p(x_t)$ (Song & Ermon, 2019; Karras et al., 2022). For plug-and-play diffusion sampling, Bayes' rule gives us:

$$\underbrace{\nabla_{x_t} \log p(x_t|y)}_{\text{Posterior Score}} = \underbrace{\nabla_{x_t} \log p(x_t)}_{\text{Prior Score}} + \underbrace{\nabla_{x_t} \log p(y|x_t)}_{\text{Likelihood Score}} \tag{1}$$

While the prior score is provided by the unconditional diffusion model, the likelihood score $\nabla_{x_t} \log p(y|x_t)$ is often intractable to compute directly, necessitating approximation strategies (Chung et al., 2023; He et al., 2023; Song et al., 2023b).

Prior work has predominantly studied improving the likelihood score approximation. Orthogonal to this, we investigate whether information from earlier sampling steps can help mitigate approxima-

tion errors in later sampling steps. Instead of developing more sophisticated approximation methods, we use adaptive moment estimation—a technique from stochastic optimization, Adam (Kingma & Ba, 2014)—to stabilize the noisy guidance gradients that arise in plug-and-play methods.

Our approach maintains exponential moving averages of the first and second moments of the likelihood score estimates across sampling steps, dampening noise while preserving the guidance signal. Despite its simplicity, this modification yields substantial improvements for DPS and CG.

We also examine how task difficulty affects performance. Typically, plug-and-play methods are evaluated under mild degradations that provide strong guidance signal (e.g., 4× super-resolution). We show as difficulty increases (e.g., 16× super-resolution), many methods degrade rapidly. We find that across all difficulties, adaptive moment estimation consistently improves the DPS baseline while other guidance methods fall below DPS alone.

Our contributions are: (i) we demonstrate that adaptive moment estimation can substantially improve plug-and-play guidance methods, achieving state-of-the-art results with minimal added complexity; (ii) we provide a variety of empirical analyses that illustrate how our method stabilizes noisy gradients and improves performance; and (iii) we reveal that task difficulty significantly impacts relative method performance, suggesting the need for more comprehensive evaluation standards.

## 2 BACKGROUND

**Diffusion Models and Score-Based Sampling.** Diffusion models (Sohl-Dickstein et al., 2015; Song & Ermon, 2019; Ho et al., 2020) learn to reverse a gradual process that adds noise to data. Starting from a clean data sample $x_0$, we repeatedly add small amounts of Gaussian noise until the result becomes almost pure noise. We describe this process using a timestep $t \in [0, T]$, where each $t$ corresponds to a noise level $\sigma_t$. The variable $\sigma_t$ measures how much total noise has been added at that step, and $x_t$ denotes the resulting noisy version of the data. As $t$ increases, the amount of noise also increases, so that $\sigma_0 < \cdots < \sigma_T$. In practice, we set $\sigma_0$ so that $x_0$ is the original data, and we choose $\sigma_T$ large enough that $x_T$ is indistinguishable from Gaussian noise. The specific schedule of noise levels $\{\sigma_t\}_{t=0}^{T}$ can vary, but it always follows this pattern of gradually increasing noise.

For two timesteps $s$ and $t$ with $s < t$, the noising process describes how a partially noised sample $x_s$ becomes a noisier sample $x_t$. This transition follows a Gaussian distribution $p(x_t|x_s) = \mathcal{N}(x_t; x_s, \sigma_{t|s}^2 \mathbb{I})$, where the variance between the two steps is given by $\sigma_{t|s}^2 = \sigma_t^2 - \sigma_s^2$. [1] Because the sum of Gaussian variables is also Gaussian, we can treat many small noising steps as one combined step. Thus, the noising process also defines a direct marginal distribution from the original data $x_0$ to any timestep $t$, $p(x_t|x_0) = \mathcal{N}(x_t; x_0, \sigma_t^2 \mathbb{I})$. Equivalently, we can express the noisy sample as $x_t = x_0 + \sigma_t \epsilon$, where $\epsilon \sim \mathcal{N}(0, \mathbb{I})$. This form shows that $x_t$ is simply the clean data $x_0$ plus Gaussian noise scaled by the noise level $\sigma_t$.

To reverse the noising process, we train a neural network $\epsilon_\theta$ to predict the noise that was added to the clean data $x_0$ in order to produce the noisy sample $x_t$. The standard training objective minimizes the difference between the true noise $\epsilon$ and the network's prediction (Ho et al., 2020):

$$\mathcal{L}_\epsilon(\theta) = \mathbb{E}_{t,x_0,\epsilon} \left[ \|\epsilon_\theta(x_t, t) - \epsilon\|_2^2 \right]. \tag{2}$$

By learning to predict this noise, the network implicitly learns the gradient of the log-probability density of the data—also known as the *score function* (Kingma & Gao, 2023):

$$s_\theta(x_t, t) = -\frac{\epsilon_\theta(x_t, t)}{\sigma_t} \approx \nabla_{x_t} \log p(x_t). \tag{3}$$

Intuitively, the added noise moves data away from regions of high likelihood. Thus, predicting the noise tells us how to move $x_t$ to increase its probability under the data distribution, which is precisely the meaning of the gradient $\nabla_{x_t} \log p(x_t)$. Once the score function is known, we can generate new data by starting from pure noise $x_T \sim \mathcal{N}(0, \sigma_T^2 \mathbb{I})$ and gradually denoising it. This sampling procedure follows *annealed Langevin dynamics* (Song & Ermon, 2019; Karras et al., 2022):

$$x_s = x_t + (\sigma_t^2 - \sigma_s^2) \nabla_{x_t} \log p(x_t) + \sqrt{\sigma_t^2 - \sigma_s^2} \frac{\sigma_s}{\sigma_t} \epsilon. \tag{4}$$

---

[1] We adapt the notation of Kingma et al. (2021) to the *variance exploding* formulation for simplicity.

This update incrementally removes noise until we obtain a clean sample from the data distribution. At any timestep $t$, the denoising network provides an estimate of the underlying clean data:

$$x_{0|t} = \mathbb{E}[x_0|x_t] = x_t - \sigma_t\,\epsilon_\theta(x_t, t), \tag{5}$$

which corresponds to the minimum mean squared error estimator under optimal training (Efron, 2011). This estimate, denoted $x_{0|t}$, plays a central role in many plug-and-play guidance methods.

**The Plug-and-Play Guidance Challenge.** The goal of plug-and-play diffusion sampling is to draw samples from the posterior distribution $p(x_0|y)$, which represents data consistent with the observation $y$. This approach relies on the decomposition of the posterior score described in Equation 1. The first term, the *prior score*, is provided by the diffusion model itself and well approximated using the trained network $\epsilon_\theta$. The second term, the *likelihood score*, expresses how the observation $y$ changes the probability of a noisy sample $x_t$. However, this term is much harder to compute in practice.

To evaluate the likelihood score, we would need to marginalize over all possible clean samples $x_0$ that could have produced $x_t$:

$$p(y|x_t) = \int p(y|x_0)\,p(x_0|x_t)\,dx_0. \tag{6}$$

This integral is generally intractable, since it requires integrating over the entire data space. Even if we train a neural network to approximate $p(y|x_t)$ directly, its gradients tend to be noisy and unstable, leading to poor guidance during sampling.

## 3 GUIDANCE WITH ADAPTIVE MOMENT ESTIMATION

Plug-and-play diffusion sampling requires approximating the often intractable likelihood score $\nabla_{x_t} \log p(y|x_t)$ to guide the sampling process toward desired conditions. In practice, this score is typically approximated by $-\nabla\mathcal{L}(x_t, y)$, where $\mathcal{L}$ is the negative log likelihood and $y$ the condition.

**Existing Likelihood Score Approximations.** There are two foundational approaches for this approximation, distinguished by the domain in which the guidance function operates. DPS (Chung et al., 2023), a widely adopted plug-and-play method, approximates the likelihood score as,

$$\nabla_{x_t} \log p(y|x_t) \approx \nabla_{x_t} \log p(y|x_{0|t}) \approx -\nabla_{x_t}\mathcal{L}(f_\phi(x_{0|t}), y) \tag{7}$$

where $x_{0|t}$ is the predicted *clean* sample from Equation 5. The DPS approximation replaces the noisy latent $x_t$ with the denoising model's estimate $x_{0|t}$, providing a point estimate of the likelihood. This allows the guidance model $f_\phi : \mathcal{X} \to \mathcal{Y}$ to be any differentiable function that operates on clean data—both learned models (e.g., pretrained neural networks) and analytical models (e.g., Gaussian blur kernels)—though it requires backpropagation through the denoising network.

Alternatively, Classifier Guidance (CG) (Dhariwal & Nichol, 2021) uses the approximation,

$$\nabla_{x_t} \log p(y|x_t) \approx -\nabla_{x_t}\mathcal{L}(f_\phi(x_t, t), y). \tag{8}$$

In contrast to DPS, CG trains a specialized guidance model $f_\phi : \mathcal{X}_t \times \mathcal{T} \to \mathcal{Y}$ that operates directly on *noisy* latents $x_t$, conditioning on the diffusion timestep $t$, making the guidance model time-aware. This eliminates the need for DPS's point estimate approximation and provides a more direct estimate of the likelihood score, though it requires training a classifier that depends on $t$.

**The Plug-and-Play Sampling Update.** We can now examine the typical plug-and-play sampling update by replacing the prior score in Equation 4 with the posterior score and decomposing it as shown in Equation 1. Finally replacing the prior score with the denoising network's approximation $s_\theta$, and the likelihood score with either of the approximations given in Equation 7 or Equation 8, we observe that the plug-and-play sampling update is performing gradient descent on the likelihood component at each timestep:

$$x_s = x_t + (\sigma_t^2 - \sigma_s^2)\Big(s_\theta(x_t, t) - \nabla\mathcal{L}(\cdot)\Big) + \sqrt{\sigma_t^2 - \sigma_s^2}\frac{\sigma_s}{\sigma_t}\epsilon. \tag{9}$$

**Stabilization via Adaptive Moments.** Regardless of the approximation strategy, guidance methods suffer from substantial noise in likelihood score estimates. In stochastic optimization, prominent

---

**Algorithm 1** Adam Diffusion Posterior Sampling (AdamDPS)

---

**Require:** Diffusion Model $\theta$, Guidance Model $f_\phi$, Condition $y$, Guidance Strength $\rho$, Sampling Timesteps $t_n, \ldots, t_0 \subseteq \mathcal{T}$, First Moment $m = \mathbf{0}$, Second Moment $v = \mathbf{0}$, Adam Step $k = 0$, First Moment Exponential Decay Rate $\beta_1$, Second Moment Exponential Decay Rate $\beta_2$

1: $x_{t_n} \sim \mathcal{N}(0, \mathbb{I})$
2: **for** $t = t_n, \ldots, t_1$ **do**
3:     $g_t = -\nabla_{x_t} \mathcal{L}(f_\phi(x_{0|t}), y)$
4:     $\hat{g}_t, m, v, k = \text{AdaptiveMomentEstimate}(g_t, m, v, k, \beta_1, \beta_2)$
5:     $x_s = \text{Sample}(x_{0|t}, x_t, t, s) + \rho \hat{g}_t$
6: **end for**
7: **return** $x_{t_0}$

---

**Algorithm 2** Adam Classifier Guidance (AdamCG)

---

**Require:** Diffusion Model $\theta$, Guidance Model $f_\phi$, Condition $y$, Guidance Strength $\rho$, Sampling Timesteps $t_n, \ldots, t_0 \subseteq \mathcal{T}$, First Moment $m = \mathbf{0}$, Second Moment $v = \mathbf{0}$, Adam Step $k = 0$, First Moment Exponential Decay Rate $\beta_1$, Second Moment Exponential Decay Rate $\beta_2$

1: $x_{t_n} \sim \mathcal{N}(0, \mathbb{I})$
2: **for** $t = t_n, \ldots, t_1$ **do**
3:     $g_t = -\nabla_{x_t} \mathcal{L}(f_\phi(x_t, t), y)$
4:     $\hat{g}_t, m, v, k = \text{AdaptiveMomentEstimate}(g_t, m, v, k, \beta_1, \beta_2)$
5:     $x_s = \text{Sample}(x_{0|t} + \rho \hat{g}_t \sigma_t^2, x_t, t, s)$
6: **end for**
7: **return** $x_{t_0}$

---

methods like RMSProp (Tieleman & Hinton, 2012) and Adam (Kingma & Ba, 2014) use adaptive moment estimation to stabilize noisy gradients. Inspired by this, we maintain exponential moving averages of the likelihood score estimates and their squared values across sampling steps,

$$m_k = \beta_1 m_{k-1} + (1 - \beta_1) g_t \qquad\qquad v_k = \beta_2 v_{k-1} + (1 - \beta_2) g_t^2 \qquad (10)$$

where $k$ is a step counter, $g_t = -\nabla \mathcal{L}(\cdot)$, and $g_t^2$ denotes element-wise squaring. The stabilized likelihood score is then computed as,

$$\hat{g}_t = \frac{\hat{m}_k}{\sqrt{\hat{v}_k} + \delta} \qquad (11)$$

with bias-corrected moments $\hat{m}_k = m_k / (1 - \beta_1^k)$ and $\hat{v}_k = v_k / (1 - \beta_2^k)$, and $\delta$ a small constant for numerical stability (Kingma & Ba, 2014).

Adaptive moment estimation serves two purposes: the first moment (momentum) smooths the optimization trajectory by accumulating gradient information across steps, while the second moment adaptively scales the updates based on the historical variance of each gradient component. Since likelihood score approximations can vary dramatically across noise levels during diffusion sampling, this stabilization is essential for improving performance. We denote the resulting methods as AdamDPS (Algorithm 1) when applied to DPS and AdamCG (Algorithm 2) when applied to CG, and demonstrate that this simple modification yields substantial improvements in sample quality.

## 4 SYNTHETIC STUDY

We begin by comparing DPS and AdamDPS in a synthetic setting, where both the true likelihood score and the ideal DPS likelihood score approximation can be computed exactly in closed form. This allows us to study the effect noise has on guidance. To this end, the data distribution is defined by a 2-dimensional Gaussian Mixture Model (GMM) with three components.

Guidance methods typically suffer from noisy likelihood score estimates due to limited conditioning information and taking gradients through large neural networks. Since our synthetic setting is free of these noise sources, we simulate them by adding to the guidance term Gaussian noise of magnitude $\zeta \|\nabla_{x_t} \log p(y|x_{0|t})\|$. In Figure 1 (Left), we measure how close the empirical distributions are to the target distribution as a function of the guidance noise coefficient $\zeta$. We find that AdamDPS is much more robust to noise than DPS, achieving lower KL divergence with the target distribution as $\zeta$ increases. Figure 1 (Right) shows this qualitatively for a specific $\zeta$.

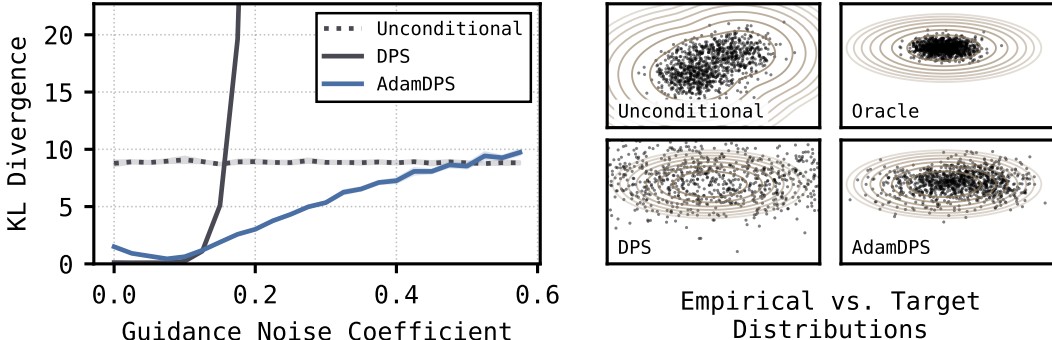

Figure 1: Left: The KL divergence between each method's empirical distribution and the target distribution as a function of the guidance noise coefficient $\zeta$. Right: Visualization of the empirical and target distributions at $\zeta = 0.175$.

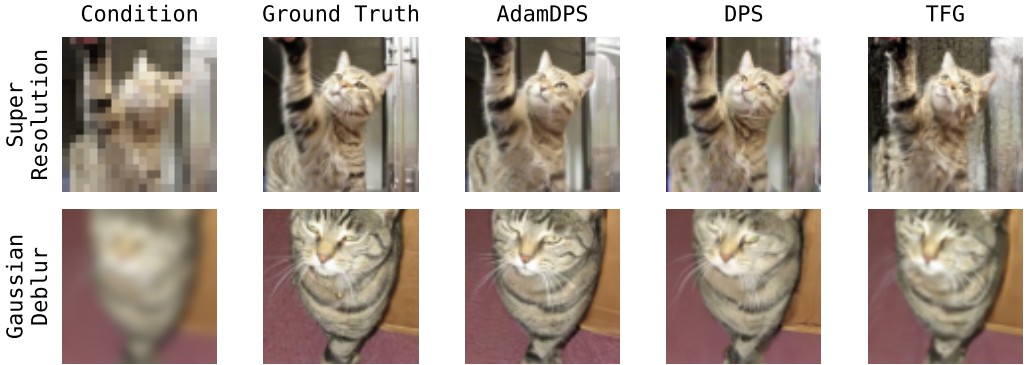

Figure 2: Qualitative comparison of AdamDPS, DPS, and TFG on Cats dataset for super resolution at 12x downsampling and Gaussian deblurring at blur intensity 9.

## 5 EXPERIMENTS

We evaluate adaptive moments for plug-and-play guidance across a range of tasks. All methods benchmarked were tuned with Bayesian Optimization (Jones et al., 1998) on a held-out validation set of 32 images. Reconstruction tasks were tuned to minimize LPIPS (Zhang et al., 2018), while class-conditional sampling was tuned for CMMD (Jayasumana et al., 2024) since tuning for accuracy encouraged generating adversarial examples. We measure alignment with the desired condition using LPIPS for reconstruction tasks, and accuracy for class-conditional sampling. Accuracy is computed as the harmonic mean across three held-out classifiers. We also report FID (Heusel et al., 2017) as a measure of fidelity, computed on 2048 samples following the evaluation procedure from Ye et al. (2024). We benchmark a variety of methods including Loss Guided Diffusion (LGD) (Song et al., 2023b), Manifold Preserving Guided Diffusion (MPGD) (He et al., 2023), Regularization by diffusion (RED-diff) (Mardani et al., 2024), DPS, UGD, and TFG on ImageNet (Deng et al., 2009), CIFAR-10 (Krizhevsky et al., 2009), and the Cats subset of the Cats vs. Dogs dataset (Elson et al., 2007). We set $N_{\text{recur}} = 1$ for TFG and sweep $N_{\text{iter}} = 1, 2, 4$ for UGD and TFG, while tuning the remaining hyperparameters as recommended by Ye et al. (2024). Additional details in Appendix A.

**Reconstruction.** Qualitatively, AdamDPS generates sharper, more realistic details in regions underdetermined by the conditioning information, where other methods tend to produce blurred or artifacted outputs. TFG reconstructions frequently exhibit visual artifacts, and DPS lacks the fine detail achieved by AdamDPS, see Figure 2. Quantitatively, for both ImageNet and the Cats dataset, AdamDPS outperforms all other methods on all reconstruction tasks: super resolution at 16x downsampling, Gaussian deblurring at blur intensity 12, and inpainting with a 90% random mask, see Figure 3. Interestingly, the second best performing method across datasets in super resolution and Gaussian deblurring is DPS. We observe this happens in challenging settings where the conditioning information does not fully determine the target image, causing other methods to degrade signif-

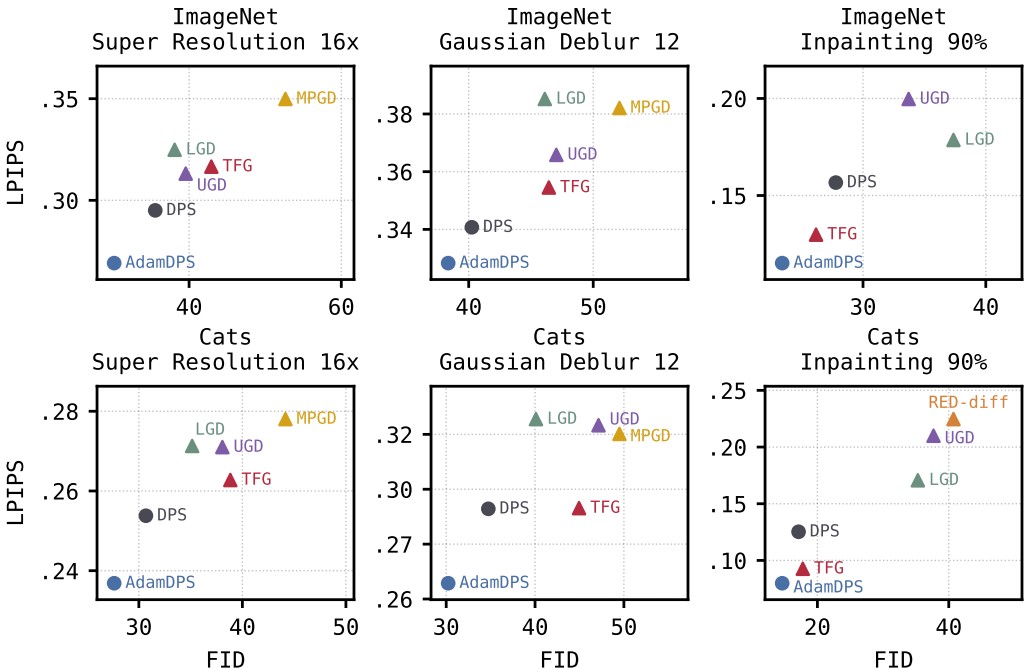

Figure 3: Reconstruction performance measured in LPIPS and FID, where lower is better for both. Comparison on ImageNet and Cats dataset for super resolution at 16x downsampling, Gaussian deblurring at blur intensity 12, and inpainting with a 90% random mask.

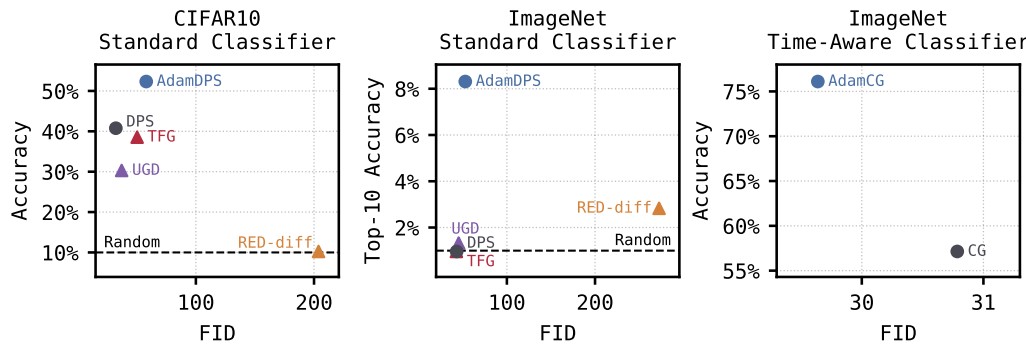

Figure 4: Class-conditional sampling performance measured in classification accuracy and FID, where higher accuracy and lower FID is better. Accuracy is computed as the harmonic mean across three held-out classifiers. Left & Center: Comparison of plug-and-play methods with a standard classifier on CIFAR-10 and ImageNet, respectively. Right: Comparison of plug-and-play methods with a time-aware classifier on ImageNet.

icantly while DPS is more robust. Inpainting proves easier than 16x super resolution and intensity 12 Gaussian deblurring, evidenced by lower LPIPS for all methods. In this setting, TFG $N_{\text{iter}} = 4$ can outperform DPS, yet still underperforms AdamDPS.

**Class-Conditional Sampling.** Adaptive moments demonstrates strong performance on class-conditional tasks, see Figure 4. On CIFAR-10, AdamDPS outperforms the next best method, DPS, by 9.86 points in classification accuracy. On ImageNet, all approaches except AdamDPS achieve top-10 classification accuracies at approximately 1%, equivalent to random chance. In contrast, AdamDPS achieves 10.49%, demonstrating substantial improvement for conditional generation in challenging settings. We also evaluate adaptive moments in the time-aware classifier guidance setting, where AdamCG improves upon CG by more than 19 points in classification accuracy.

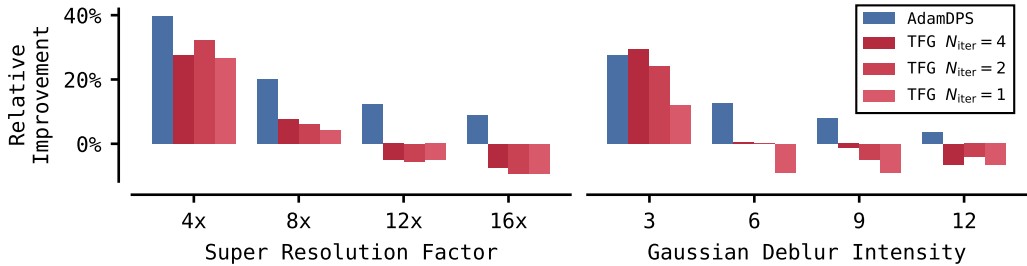

Figure 5: Relative improvement over DPS on ImageNet as task difficulty increases for super resolution and Gaussian deblurring.

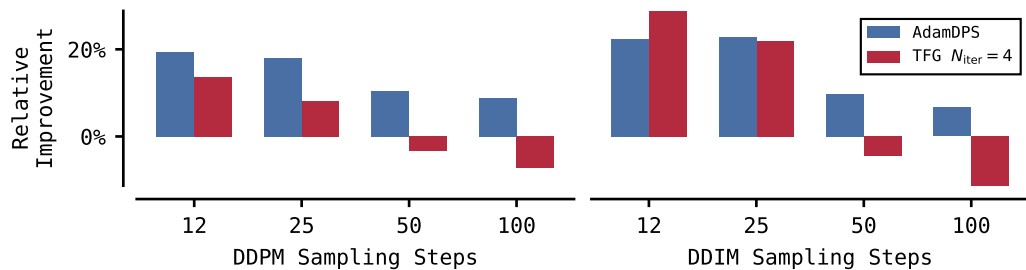

Figure 6: Relative improvement over DPS for super resolution at 16x downsampling on ImageNet, across DDPM and DDIM sampling step budgets.

**Task Difficulty Ablation.** We examine how task difficulty affects the relative performance of AdamDPS and TFG compared to DPS. Figure 5 shows the relative LPIPS improvement of AdamDPS and TFG over DPS on ImageNet as super resolution and deblurring tasks increase in difficulty. For super resolution, AdamDPS consistently outperforms TFG, maintaining a substantial improvement over DPS across all difficulty levels. TFG only improves upon DPS in the two easiest settings, and performs noticeably worse in the two hardest. We see a similar trend for Gaussian deblurring. TFG only meaningfully improves upon DPS in the easiest setting, performing equivalently to or worse than DPS as difficulty increases. AdamDPS demonstrates robust positive improvement that becomes more pronounced at higher blur intensities, as TFG variants show declining performance and ultimately underperform DPS on the most challenging tasks. AdamDPS remains effective even as conditioning information becomes increasingly limited.

**Sampling Steps Ablation.** We analyze the effect of reducing the sampling step budget on AdamDPS and TFG relative to DPS for super resolution at 16x downsampling on ImageNet, see Figure 6. For both DDPM and DDIM sampling, AdamDPS consistently improves over DPS across all step counts. TFG $N_{\text{iter}} = 4$ fails to outperform at high step counts, consistent with Figure 3. However, TFG becomes competitive at lower step counts, possibly because TFG performs $1 + N_{\text{iter}}$ updates to the image per step, whereas DPS and AdamDPS only perform 1 update per step. Across both samplers, AdamDPS provides reliable gains over DPS regardless of step budget.

**Wall Clock.** Figure 7 shows wall clock time averaged over 5 trials for ImageNet class-conditional sampling with 100 steps on a single H100 GPU, generating a batch of 8 256×256 images. AdamDPS adds negligible overhead compared to DPS. Both AdamDPS and DPS outpace TFG, whose gradient computations scale with $N_{\text{recur}}(1 + N_{\text{iter}})$ through the guidance model and $N_{\text{recur}}$ through the denoising network.

**Adam $\beta_1$, $\beta_2$ Ablation.** Figure 7 shows an ablation of Adam hyperparameters $\beta_1$ and $\beta_2$ for AdamDPS across super resolution at 16x downsampling, Gaussian deblurring at blur intensity 12, and inpainting with a 90% random mask. We compare the relative LPIPS improvement over DPS for three configurations: default AdamDPS, AdamDPS with $\beta_1 = 0$, and AdamDPS with $\beta_2 = 0$. AdamDPS consistently outperforms both ablated variants across all tasks, demonstrating that both momentum and adaptive scaling are essential for optimal performance, with their relative importance varying by task.

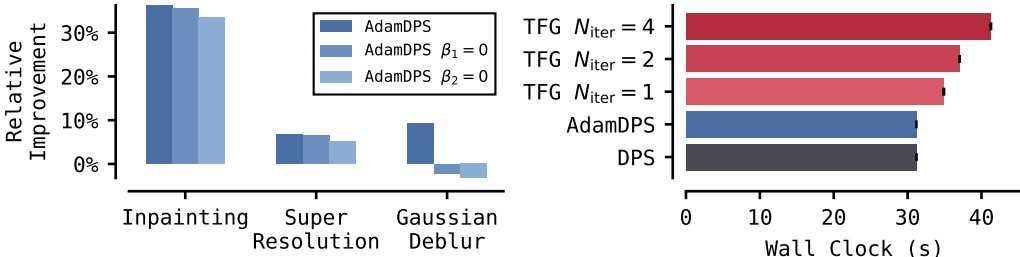

Figure 7: Left: Ablation of Adam $\beta_1$, $\beta_2$ for super resolution at 16x downsampling, Gaussian deblurring at blur intensity 12, and inpainting at 90% random mask on the Cats dataset. Right: Wall clock comparison on a single H100 GPU of 100 step class-conditional sampling with a standard classifier on ImageNet for a batch of 8 256x256 images.

**Sampling Analysis.** We visualize the sampling trajectories of DPS and AdamDPS for super resolution at 16x downsampling on ImageNet in Figure 8. We project the trajectories onto two dimensions of interest: the first defined by the difference between the initial noise and the target image, and the second defined by the difference between the AdamDPS and DPS solutions. The contours indicate the Mean Squared Error (MSE) loss surface with respect to the target. Across samples, AdamDPS trajectories more directly approach the ground truth, while DPS trajectories may stray from the target, resulting in solutions less aligned with the condition.

To better understand the performance gap between AdamDPS and DPS, we examine the cosine similarity between sequential guidance terms $g_t$ and $g_s$ throughout sampling for both methods. As shown in Figure 9, DPS guidance terms in adjacent steps frequently disagree in direction, with negative cosine similarity for the majority of sampling. This suggests that DPS guidance is often fighting itself, pushing the sample in conflicting directions across consecutive steps. In contrast, AdamDPS maintains consistently positive cosine similarity between adjacent guidance terms, indicating that its

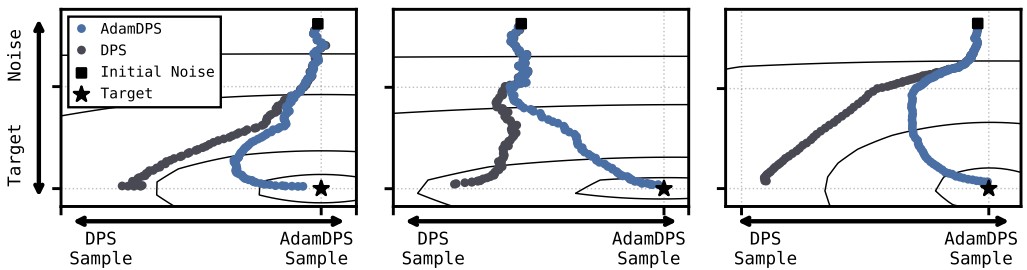

Figure 8: Sampling trajectories for DPS and AdamDPS projected onto two dimensions for super resolution at 16x downsampling on ImageNet. The y-axis is defined by the difference between the initial noise and the target, and the x-axis by the difference between the AdamDPS and DPS solutions. Contours depict the MSE loss surface with respect to the target.

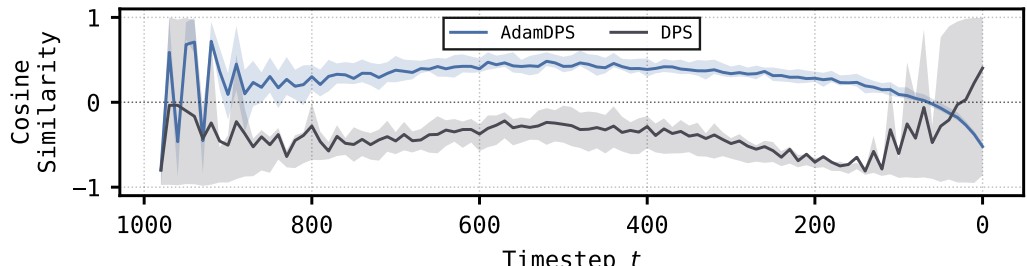

Figure 9: Cosine similarity between sequential guidance terms $g_t$ and $g_s$ throughout sampling for DPS and AdamDPS. Guidance terms collected from 16x super resolution task on ImageNet. Shading denotes the 25th to 75th percentile.

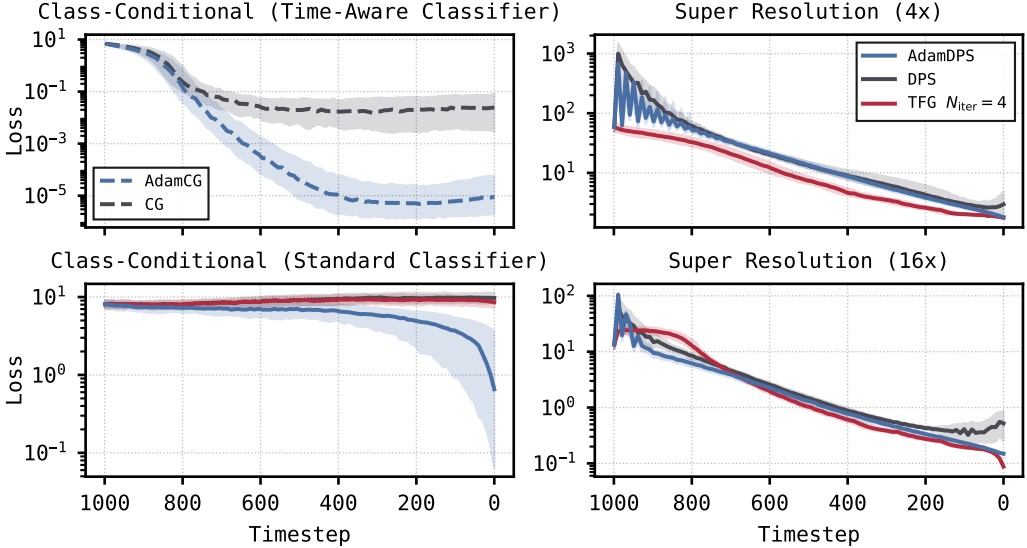

Figure 10: The guidance loss throughout sampling for class-conditional and super resolution tasks on ImageNet. Shading denotes the 25th to 75th percentile.

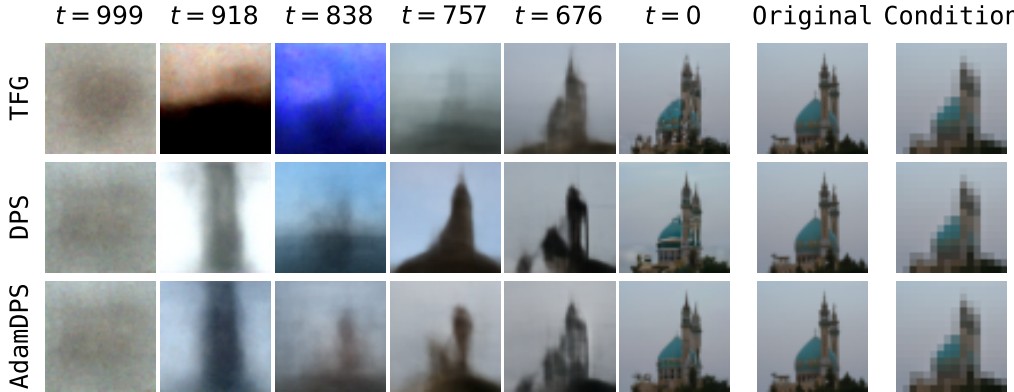

Figure 11: Qualitative example of sampling trajectory for super resolution at 16x downsampling on ImageNet. Displayed are the intermediate clean data predictions $x_{0|t}$ throughout sampling.

updates agree in direction for most of sampling. This coherence allows AdamDPS to make steady progress towards aligning with the condition rather than oscillating unproductively.

We track the loss $\mathcal{L}(f_\phi(\cdot), y)$ along the sampling trajectory to compare how each method minimizes the guidance objective, see Figure 10. For 4x super resolution on ImageNet, where conditioning information is relatively dense, TFG reduces loss more quickly than DPS and AdamDPS initially, though all methods achieve similar terminal loss values. Despite comparable final losses, AdamDPS still slightly outperforms TFG in reconstruction quality. For 16x super resolution on ImageNet, TFG struggles to reduce loss early in sampling, only achieving a rapid decrease near the end. Importantly, this lower terminal loss compared to DPS and AdamDPS does not translate to better reconstruction performance, as evident in Figure 3. When conditioning information is severely limited, over-optimizing to the sparse signal can be detrimental rather than beneficial. In this case, TFG effectively overfits to the low-resolution image, producing the visual artifacts observed in Figure 2.

For class-conditional sampling on ImageNet with a standard classifier—a particularly challenging setting where baselines achieve near-random accuracy—neither DPS nor TFG can meaningfully reduce the guidance loss, as shown in Figure 10. In contrast, AdamDPS achieves slow initial progress that accelerates toward the end of sampling, successfully reducing the guidance loss when all other methods could not, consistent with its accuracy improvement in Figure 4. For time-aware classifiers,

both CG and AdamCG reduce loss throughout sampling, but AdamCG continues improving while CG plateaus, ultimately achieving substantially lower loss.

We visualize intermediate clean data predictions $x_{0|t}$ during the early stages of sampling for 16x super resolution on ImageNet in Figure 11. DPS and AdamDPS produce reasonable predictions of the target image from the earliest timesteps, with structure and color emerging quickly. In contrast, TFG's predictions vary wildly and bear little resemblance to either the target image or conditioning image. This is consistent with the guidance loss curves in Figure 10, where TFG fails to reduce loss early in sampling for 16x super resolution. These erratic early predictions further suggest that TFG's guidance is ineffective when conditioning information is sparse, leaving the model unable to identify a coherent trajectory toward aligning with the condition. Moreover, TFG's final sample exhibits visual artifacts despite its aggressive loss reduction late in sampling, illustrating how overfitting to sparse conditioning information degrades reconstruction quality.

Additional quantitative and qualitative results are provided in Appendix C and Appendix D.

## 6 RELATED WORK

Beyond DPS and CG, numerous methods have been proposed to improve conditional generation through more sophisticated likelihood score approximations and algorithmic refinements. MPGD (He et al., 2023) sidesteps backpropagation through the diffusion model by optimizing directly in data space on the clean data estimate $x_{0|t}$, while LGD (Song et al., 2023b) applies Monte Carlo smoothing to stabilize the DPS likelihood score approximation. Building upon these approaches, compositional frameworks combine multiple guidance strategies. UGD (Bansal et al., 2023) unifies DPS and MPGD with recurrence, introducing separate hyperparameters to control the contribution of different components. Recurrence is a key technique employed by guidance methods such as FreeDoM (Yu et al., 2023), which enables iterative refinement by revisiting timesteps (Mokady et al., 2023; Wang et al., 2022; Lugmayr et al., 2022; Du et al., 2023). TFG (Ye et al., 2024) further extends this by combining DPS, MPGD, LGD, UGD, and FreeDoM into a unified framework. TFG introduces hyperparameters for various algorithmic components, along with schedules that adjust some of these hyperparameters throughout sampling.

Additional methods exist which leverage specific structural assumptions about the guidance model to achieve improved performance. DDRM (Kawar et al., 2022) assumes linearity of the inverse problem for efficient posterior sampling through variational inference, while ΠGDM (Song et al., 2023a) handles non-differentiable measurements by assuming the availability of a pseudoinverse. TMPD (Boys et al., 2023) and FreeHunch (Rissanen et al., 2025) assume access to linear measurement operators to estimate denoiser covariance, yielding improved likelihood score approximations.

In contrast to these approaches, which focus on improving the likelihood score approximation itself, we focus on reducing the noise in guidance updates that arises from approximate likelihood scores and limited conditioning information. Our work is orthogonal to these plug-and-play methods; adaptive moment estimation can be applied to any of these methods to stabilize guidance updates. In concurrent work, Gallon et al. (2026) apply Adam to DPS, as part of their spectral diffusion framework for solving PDEs, and observe improvements over DPS alone.

## 7 CONCLUSION

We demonstrate that adaptive moment estimation, which stabilizes noisy guidance during sampling, can substantially improve plug-and-play diffusion sampling. AdamDPS and AdamCG achieve state-of-the-art performance across diverse reconstruction and class-conditional generation tasks with minimal computational overhead. Adding adaptive moments is typically only a few-line modification to an existing guidance implementation, and we show in Appendix C that it extends beyond DPS and CG, yielding improvements when applied to other plug-and-play guidance methods. Importantly, our evaluation across a range of task difficulties reveals that complex methods like TFG which outperform simpler baselines on easy tasks often degrade substantially as conditioning information becomes limited, ultimately underperforming DPS. Noise mitigation offers a simple yet effective alternative to increasingly sophisticated likelihood score approximations.

ACKNOWLEDGMENTS

We thank Chia-Hao Lee and David A. Muller for productive discussions which motivated this work. CB is supported by the National Science Foundation (NSF) through the NSF Research Traineeship (NRT) program under Grant No. 2345579. JL is supported by a Google PhD Fellowship. This work is also supported by the NSF through Grant No. OAC-2118310 and the AI Research Institutes program Award No. DMR-2433348; the National Institute of Food and Agriculture (USDA/NIFA); the Air Force Office of Scientific Research (AFOSR); New York-Presbyterian for the NYP-Cornell Cardiovascular AI Collaboration; and a Schmidt AI2050 Senior Fellowship.

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

## A  ADDITIONAL EXPERIMENTAL DETAILS

We build on the TFG codebase[2] using their implementations for CG, MPGD, UGD, and TFG. We slightly update the LGD and DPS implementations. For the TFG baseline itself we follow the authors' recommendation, tuning $\rho, \mu, \gamma$, sweeping $N_{\text{iter}} = 1, 2, 4$, setting $N_{\text{recur}} = 1$, and setting the schedules for $\rho, \mu, \gamma$ to be increasing, increasing, decreasing (Ye et al., 2024). All experiments use 100 step DDPM sampling unless stated otherwise. Gaussian deblurring blur intensity refers to the standard deviation of the Gaussian kernel used to blur the image. We split the datasets into validation and test splits and tune our hyperparameters on the validation split. Our validation and test splits contain 32 and 2048 images respectively. We use Bayesian Optimization to tune the hyperparameters of each method on our validation split. We perform 150 trials, the first 50 trials select candidates by Sobol sampling, the remaining 100 trials use Log Noisy Expected Improvement (Ament et al., 2023). Accuracy on class-conditional sampling tasks is computed with three held-out classifiers for each dataset downloaded from HuggingFace. For CIFAR10: ViT[3], ViT Finetuned[4], and ConvNeXt[5]. For ImageNet: ViT[6], DINO v2[7], MobileViT v2[8].

## B  ADDITIONAL PSEUDOCODE

---

**Algorithm 3** Diffusion Posterior Sampling (DPS)

---

**Require:** Diffusion Model $\theta$, Guidance Model $f_\phi$, Condition $y$, Guidance Strength $\rho_{t_n}, \ldots, \rho_{t_0}$, Sampling Timesteps $t_n, \ldots, t_0 \subseteq \mathcal{T}$
1: $x_{t_n} \sim \mathcal{N}(0, \mathbb{I})$
2: **for** $t = t_n, \ldots, t_1$ **do**
3: $\quad g_t = -\rho_t \nabla_{x_t} \mathcal{L}(f_\phi(x_{0|t}), y)$
4: $\quad x_s = \text{Sample}(x_{0|t}, x_t, t, s) + g_t$
5: **end for**
6: **return** $x_{t_0}$

---

**Algorithm 4** Classifier Guidance (CG)

---

**Require:** Diffusion Model $\theta$, Guidance Model $f_\phi$, Condition $y$, Guidance Strength $\rho$, Sampling Timesteps $t_n, \ldots, t_0 \subseteq \mathcal{T}$
1: $x_{t_n} \sim \mathcal{N}(0, \mathbb{I})$
2: **for** $t = t_n, \ldots, t_1$ **do**
3: $\quad g_t = -\nabla_{x_t} \mathcal{L}(f_\phi(x_t, t), y)$
4: $\quad x_s = \text{Sample}(x_{0|t} + \rho g_t \sigma_t^2, x_t, t, s)$
5: **end for**
6: **return** $x_{t_0}$

---

**Algorithm 5** Adaptive Moment Estimate

---

**Require:** Gradient $g$, First Moment Estimate $m$, Second Moment Estimate $v$, Adam Step $k$, First Moment Exponential Decay Rate $\beta_1$, Second Moment Exponential Decay Rate $\beta_2$, $\delta = 10^{-8}$
1: $k = k + 1$
2: $m = \beta_1 m + (1 - \beta_1)g$
3: $v = \beta_2 v + (1 - \beta_2)g^2$
4: $\hat{m} = m/(1 - \beta_1^k)$
5: $\hat{v} = v/(1 - \beta_2^k)$
6: $\hat{g} = \hat{m}/(\sqrt{\hat{v}} + \delta)$
7: **return** $\hat{g}, m, v, k$

---

[2] https://github.com/YWolfeee/Training-Free-Guidance
[3] https://huggingface.co/nateraw/vit-base-patch16-224-cifar10
[4] https://huggingface.co/aaraki/vit-base-patch16-224-in21k-finetuned-cifar10
[5] https://huggingface.co/ahsanjavid/convnext-tiny-finetuned-cifar10
[6] https://huggingface.co/anonymous-429/osf-vit-base-patch16-224-imagenet
[7] https://huggingface.co/facebook/dinov2-giant-imagenet1k-1-layer
[8] https://huggingface.co/apple/mobilevitv2-1.0-imagenet1k-256

# C ADDITIONAL QUANTITATIVE RESULTS

Table 1: Reconstruction on ImageNet

| Method | Inpainting | | | Super Resolution 4x | | | Super Resolution 16x | | | Gaussian Deblur 3 | | | Gaussian Deblur 12 | | |
|---|---|---|---|---|---|---|---|---|---|---|---|---|---|---|---|
| | LPIPS | FID | IS | LPIPS | FID | IS | LPIPS | FID | IS | LPIPS | FID | IS | LPIPS | FID | IS |
| $\Pi$GDM | 0.30 | 66.83 | 36.85 | 0.48 | 63.73 | 26.34 | 0.48 | 49.41 | 30.11 | 0.48 | 65.04 | 24.69 | 0.50 | 51.97 | 27.76 |
| RED-diff | 0.31 | 75.86 | 36.60 | 0.20 | 30.42 | 102.09 | 0.57 | 100.30 | 9.66 | 0.27 | 40.13 | 76.44 | 0.52 | 209.90 | 5.85 |
| DPS | 0.16 | 27.79 | 132.32 | 0.19 | 26.43 | 128.03 | 0.30 | 35.55 | 60.02 | 0.23 | 27.66 | 109.29 | 0.34 | 40.25 | 45.07 |
| LGD | 0.18 | 37.37 | 89.52 | 0.24 | 31.97 | 105.63 | 0.32 | 38.10 | 49.85 | 0.31 | 37.02 | 74.39 | 0.39 | 46.11 | 36.46 |
| MPGD | 0.45 | 142.47 | 14.47 | 0.16 | 27.75 | 120.52 | 0.35 | 52.66 | 35.93 | 0.23 | 31.18 | 107.18 | 0.38 | 52.11 | 34.34 |
| UGD$_{N_{\text{iter}}=1}$ | 0.29 | 47.30 | 60.38 | 0.31 | 38.42 | 70.41 | 0.34 | 38.52 | 45.25 | 0.33 | 37.46 | 63.02 | 0.40 | 43.06 | 36.76 |
| UGD$_{N_{\text{iter}}=2}$ | 0.20 | 33.18 | 95.94 | 0.18 | 28.31 | 123.13 | 0.32 | 40.05 | 49.68 | 0.26 | 32.76 | 98.00 | 0.38 | 44.48 | 40.11 |
| UGD$_{N_{\text{iter}}=4}$ | 0.20 | 33.72 | 95.57 | 0.15 | 25.68 | 139.34 | 0.31 | 39.56 | 49.09 | 0.23 | 29.51 | 106.57 | 0.37 | 47.04 | 38.47 |
| TFG$_{N_{\text{iter}}=1}$ | 0.20 | 41.49 | 81.77 | 0.14 | 25.37 | 132.58 | 0.32 | 42.27 | 46.69 | 0.20 | 27.21 | 113.49 | 0.36 | 47.34 | 36.11 |
| TFG$_{N_{\text{iter}}=2}$ | 0.13 | 26.17 | 135.12 | 0.13 | 25.03 | 136.73 | 0.32 | 42.69 | 46.63 | 0.17 | 26.01 | 125.65 | 0.35 | 46.44 | 38.92 |
| TFG$_{N_{\text{iter}}=4}$ | 0.13 | 26.87 | 125.08 | 0.14 | 26.62 | 128.90 | 0.32 | 42.92 | 45.20 | 0.16 | 25.74 | 128.00 | 0.36 | 49.55 | 38.74 |
| AdamMPGD | 0.48 | 142.78 | 13.39 | 0.15 | 27.06 | 120.36 | 0.35 | 56.51 | 32.74 | 0.20 | 29.96 | 114.47 | 0.37 | 51.43 | 35.29 |
| Adam$\Pi$GDM | 0.11 | 22.25 | 163.23 | 0.14 | 22.42 | 161.15 | 0.28 | 33.75 | 61.26 | 0.20 | 27.30 | 132.52 | 0.30 | 35.91 | 59.00 |
| AdamDPS | 0.12 | 23.42 | 160.63 | 0.12 | 20.92 | 180.91 | 0.27 | 30.16 | 65.69 | 0.17 | 23.45 | 144.23 | 0.33 | 38.35 | 49.11 |

Table 2: Reconstruction on ImageNet (Intermediate Difficulties)

| Method | Super Resolution 8x | | | Super Resolution 12x | | | Gaussian Deblur 6 | | | Gaussian Deblur 9 | | |
|---|---|---|---|---|---|---|---|---|---|---|---|---|
| | LPIPS | FID | IS | LPIPS | FID | IS | LPIPS | FID | IS | LPIPS | FID | IS |
| DPS | 0.23 | 29.02 | 88.64 | 0.26 | 32.63 | 71.02 | 0.26 | 31.39 | 71.41 | 0.30 | 36.65 | 54.54 |
| TFG$_{N_{\text{iter}}=1}$ | 0.22 | 31.76 | 79.87 | 0.27 | 41.54 | 52.95 | 0.29 | 35.43 | 63.57 | 0.33 | 42.91 | 46.36 |
| TFG$_{N_{\text{iter}}=2}$ | 0.22 | 31.56 | 80.09 | 0.27 | 37.08 | 57.60 | 0.26 | 33.21 | 72.71 | 0.32 | 40.15 | 49.85 |
| TFG$_{N_{\text{iter}}=4}$ | 0.22 | 32.26 | 76.58 | 0.27 | 39.84 | 53.69 | 0.26 | 33.58 | 73.29 | 0.31 | 39.58 | 49.63 |
| AdamDPS | 0.19 | 24.36 | 117.35 | 0.23 | 27.60 | 85.87 | 0.23 | 27.26 | 90.08 | 0.28 | 33.63 | 62.14 |

Table 3: Reconstruction on Cats

| Method | Inpainting | | Super Resolution 4x | | Super Resolution 16x | | Gaussian Deblur 3 | | Gaussian Deblur 12 | |
|---|---|---|---|---|---|---|---|---|---|---|
| | LPIPS | FID | LPIPS | FID | LPIPS | FID | LPIPS | FID | LPIPS | FID |
| $\Pi$GDM | 0.32 | 80.77 | 0.40 | 60.26 | 0.44 | 52.76 | 0.42 | 58.80 | 0.46 | 58.48 |
| RED-diff | 0.22 | 40.72 | 0.12 | 22.04 | 0.35 | 103.73 | 0.16 | 32.82 | 0.40 | 120.05 |
| DPS | 0.13 | 17.13 | 0.14 | 17.74 | 0.25 | 30.68 | 0.18 | 23.58 | 0.29 | 34.74 |
| LGD | 0.17 | 35.29 | 0.14 | 19.63 | 0.27 | 35.15 | 0.20 | 25.72 | 0.33 | 40.10 |
| MPGD | 0.42 | 76.60 | 0.09 | 15.70 | 0.28 | 44.16 | 0.14 | 20.88 | 0.32 | 49.50 |
| UGD$_{N_{\text{iter}}=1}$ | 0.32 | 64.22 | 0.25 | 38.53 | 0.29 | 33.60 | 0.26 | 35.70 | 0.35 | 43.10 |
| UGD$_{N_{\text{iter}}=2}$ | 0.22 | 38.14 | 0.12 | 19.86 | 0.27 | 38.07 | 0.19 | 26.97 | 0.34 | 49.98 |
| UGD$_{N_{\text{iter}}=4}$ | 0.21 | 37.68 | 0.11 | 18.56 | 0.27 | 38.07 | 0.17 | 23.91 | 0.32 | 47.14 |
| TFG$_{N_{\text{iter}}=1}$ | 0.15 | 18.24 | 0.09 | 14.81 | 0.27 | 39.75 | 0.15 | 20.78 | 0.32 | 50.49 |
| TFG$_{N_{\text{iter}}=2}$ | 0.09 | 17.78 | 0.08 | 13.54 | 0.26 | 38.82 | 0.12 | 18.20 | 0.31 | 46.90 |
| TFG$_{N_{\text{iter}}=4}$ | 0.09 | 17.77 | 0.08 | 14.09 | 0.26 | 38.83 | 0.11 | 16.42 | 0.29 | 44.95 |
| AdamMPGD | 0.44 | 70.93 | 0.10 | 15.32 | 0.28 | 46.85 | 0.12 | 18.33 | 0.30 | 49.17 |
| Adam$\Pi$GDM | 0.08 | 14.54 | 0.09 | 13.82 | 0.24 | 26.51 | 0.13 | 17.79 | 0.25 | 29.94 |
| AdamDPS | 0.08 | 14.64 | 0.09 | 13.19 | 0.24 | 27.62 | 0.14 | 20.43 | 0.27 | 30.22 |

Table 4: Reconstruction on Cats (Intermediate Difficulties)

| Method | Super Resolution 8x | | Super Resolution 12x | | Gaussian Deblur 6 | | Gaussian Deblur 9 | |
|---|---|---|---|---|---|---|---|---|
| | LPIPS | FID | LPIPS | FID | LPIPS | FID | LPIPS | FID |
| DPS | 0.21 | 25.36 | 0.22 | 25.22 | 0.22 | 26.99 | 0.26 | 31.44 |
| TFG$_{N_{\text{iter}}=1}$ | 0.17 | 23.70 | 0.22 | 31.46 | 0.24 | 31.36 | 0.27 | 38.37 |
| TFG$_{N_{\text{iter}}=2}$ | 0.16 | 22.10 | 0.22 | 29.77 | 0.21 | 27.40 | 0.27 | 36.95 |
| TFG$_{N_{\text{iter}}=4}$ | 0.17 | 23.18 | 0.21 | 29.02 | 0.20 | 26.25 | 0.25 | 33.75 |
| AdamDPS | 0.16 | 19.45 | 0.20 | 22.64 | 0.20 | 24.32 | 0.23 | 26.48 |

Table 5: Sampling Step Ablation for Super Resolution at 16x Downsampling on ImageNet

| | DDPM | | | | | | | | DDIM | | | | | | | |
| | LPIPS | | | | FID | | | | LPIPS | | | | FID | | | |
| Method | 100 | 50 | 25 | 12 | 100 | 50 | 25 | 12 | 100 | 50 | 25 | 12 | 100 | 50 | 25 | 12 |
|---|---|---|---|---|---|---|---|---|---|---|---|---|---|---|---|---|
| DPS | 0.30 | 0.33 | 0.42 | 0.58 | 35.55 | 44.54 | 62.21 | 154.29 | 0.31 | 0.34 | 0.45 | 0.57 | 41.50 | 43.62 | 51.34 | 58.01 |
| $\text{TFG}_{N_{\text{iter}}=4}$ | 0.32 | 0.34 | 0.39 | 0.50 | 42.92 | 44.55 | 52.35 | 118.59 | 0.35 | 0.35 | 0.35 | 0.41 | 58.38 | 59.20 | 53.84 | 60.34 |
| AdamDPS | 0.27 | 0.30 | 0.35 | 0.46 | 30.16 | 35.63 | 49.05 | 98.71 | 0.29 | 0.30 | 0.35 | 0.44 | 39.60 | 37.18 | 46.35 | 77.33 |

Table 6: Sampling Step Ablation for Super Resolution at 16x Downsampling on Cats

| | DDPM | | | | | | | | DDIM | | | | | | | |
| | LPIPS | | | | FID | | | | LPIPS | | | | FID | | | |
| Method | 100 | 50 | 25 | 12 | 100 | 50 | 25 | 12 | 100 | 50 | 25 | 12 | 100 | 50 | 25 | 12 |
|---|---|---|---|---|---|---|---|---|---|---|---|---|---|---|---|---|
| DPS | 0.25 | 0.29 | 0.35 | 0.52 | 30.68 | 40.15 | 58.25 | 184.52 | 0.29 | 0.29 | 0.37 | 0.51 | 30.31 | 32.35 | 42.73 | 67.90 |
| $\text{TFG}_{N_{\text{iter}}=4}$ | 0.26 | 0.29 | 0.34 | 0.40 | 38.83 | 39.72 | 44.83 | 70.17 | 0.35 | 0.34 | 0.33 | 0.34 | 55.81 | 51.70 | 49.85 | 47.33 |
| AdamDPS | 0.24 | 0.26 | 0.29 | 0.38 | 27.62 | 30.32 | 34.15 | 72.19 | 0.27 | 0.27 | 0.29 | 0.37 | 27.53 | 27.20 | 32.51 | 61.72 |

Table 7: Class-Conditional (Standard Classifier) on CIFAR10

| | Guidance Classifier Accuracy (%) | Held-out Classifier Accuracy (%) | | | FID |
| Method | | ViT | ViT Finetuned | ConvNeXt | |
|---|---|---|---|---|---|
| RED-diff | 10.7 | 9.9 | 10.2 | 10.7 | 203.8 |
| DPS | 42.8 | 39.4 | 40.3 | 42.8 | 32.3 |
| LGD | 21.8 | 19.6 | 18.1 | 21.8 | 31.0 |
| MPGD | 26.7 | 24.9 | 24.3 | 26.7 | 34.1 |
| $\text{UGD}_{N_{\text{iter}}=1}$ | 9.3 | 9.7 | 10.2 | 9.3 | 25.4 |
| $\text{UGD}_{N_{\text{iter}}=2}$ | 24.6 | 22.9 | 23.1 | 24.6 | 32.4 |
| $\text{UGD}_{N_{\text{iter}}=4}$ | 31.9 | 28.9 | 30.3 | 31.9 | 37.2 |
| $\text{TFG}_{N_{\text{iter}}=1}$ | 39.4 | 37.5 | 38.8 | 39.4 | 50.3 |
| $\text{TFG}_{N_{\text{iter}}=2}$ | 17.0 | 15.4 | 14.7 | 17.0 | 25.1 |
| $\text{TFG}_{N_{\text{iter}}=4}$ | 19.4 | 17.1 | 16.7 | 19.4 | 26.1 |
| AdamMPGD | 20.7 | 20.3 | 20.2 | 20.7 | 29.9 |
| AdamDPS | 52.6 | 53.2 | 51.2 | 52.6 | 58.0 |

Table 8: Class-Conditional (Standard Classifier) on ImageNet

| | Guidance Classifier Top-10 Accuracy (%) | Held-out Classifier Top-10 Accuracy (%) | | | FID |
| Method | | ViT | DINO v2 | MobileViT v2 | |
|---|---|---|---|---|---|
| RED-diff | 12.7 | 23.0 | 3.2 | 1.4 | 272.7 |
| DPS | 0.7 | 0.8 | 1.2 | 1.0 | 43.0 |
| LGD | 0.7 | 0.7 | 1.2 | 1.1 | 42.5 |
| MPGD | 0.9 | 0.8 | 1.2 | 1.4 | 43.6 |
| $\text{UGD}_{N_{\text{iter}}=1}$ | 0.8 | 0.6 | 0.7 | 0.7 | 42.3 |
| $\text{UGD}_{N_{\text{iter}}=2}$ | 1.4 | 1.1 | 1.4 | 1.6 | 45.3 |
| $\text{UGD}_{N_{\text{iter}}=4}$ | 1.7 | 1.0 | 0.8 | 0.9 | 43.6 |
| $\text{TFG}_{N_{\text{iter}}=1}$ | 1.0 | 0.9 | 0.9 | 1.1 | 42.7 |
| $\text{TFG}_{N_{\text{iter}}=2}$ | 0.9 | 0.8 | 0.6 | 1.1 | 43.2 |
| $\text{TFG}_{N_{\text{iter}}=4}$ | 1.4 | 0.8 | 0.9 | 1.0 | 43.1 |
| AdamMPGD | 1.4 | 1.5 | 1.6 | 1.8 | 45.8 |
| AdamDPS | 10.5 | 9.2 | 8.3 | 7.6 | 52.8 |

Table 9: Class-Conditional (Time-Aware Classifier) on ImageNet

| | Guidance Classifier Accuracy (%) | Held-out Classifier Accuracy (%) | | | FID |
| Method | | ViT | DINO v2 | MobileViT v2 | |
|---|---|---|---|---|---|
| CG | 61.0 | 63.4 | 58.3 | 51.0 | 30.8 |
| AdamCG | 82.5 | 84.2 | 77.8 | 68.1 | 29.6 |

# D    ADDITIONAL QUALITATIVE RESULTS

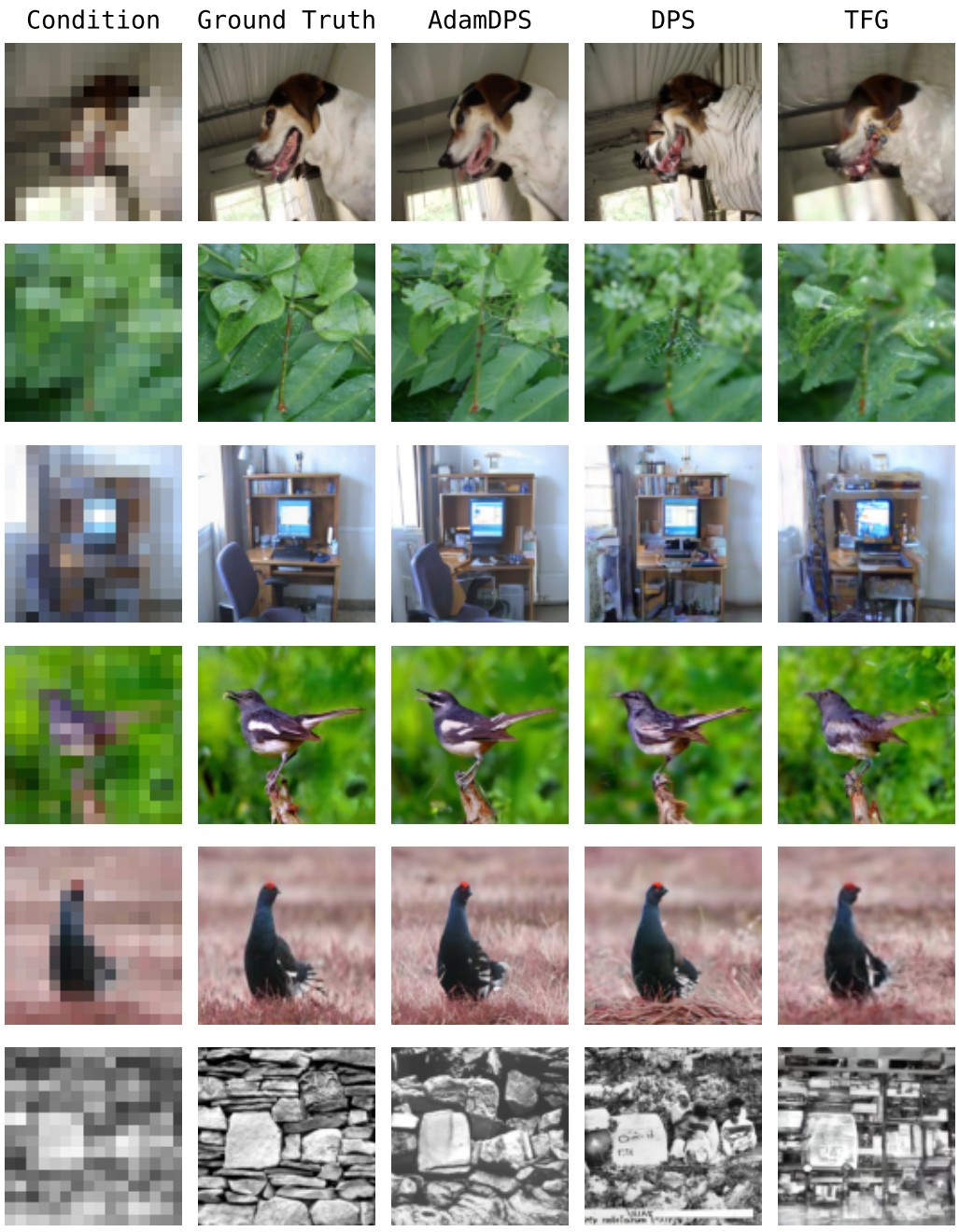

Figure 12: Additional qualitative results for super resolution at 16x downsampling on ImageNet.

Gaussian Deblur (12) on ImageNet

Figure 13: Additional qualitative results for Gaussian deblurring at blur intensity 12 on ImageNet.

Figure 14: Additional qualitative results for super resolution at 12x downsampling on Cats.

Gaussian Deblur (9) on Cats

Figure 15: Additional qualitative results for Gaussian deblurring at blur intensity 9 on Cats.

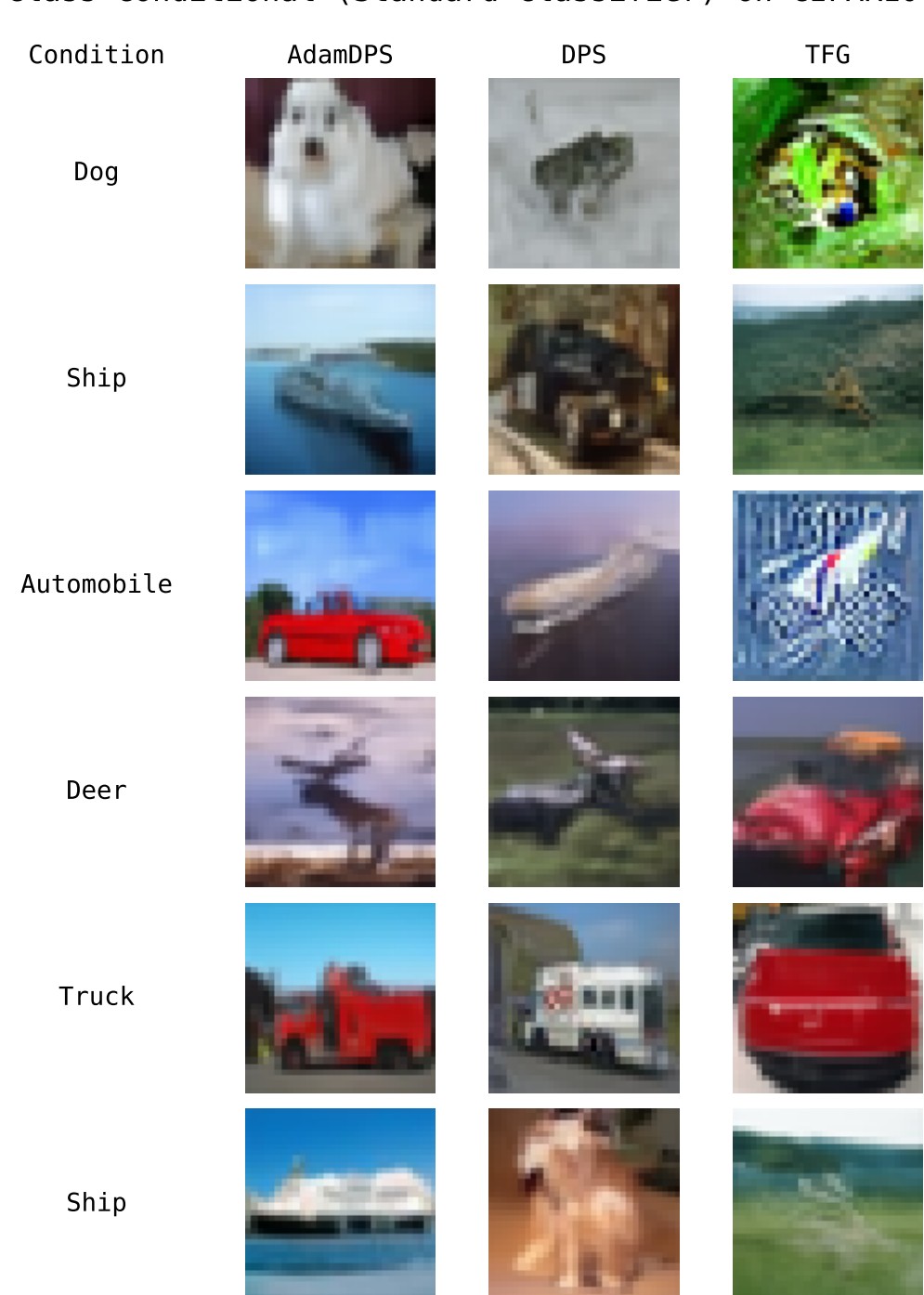

Figure 16: Additional qualitative results for class-conditional sampling with a standard classifier on CIFAR10.

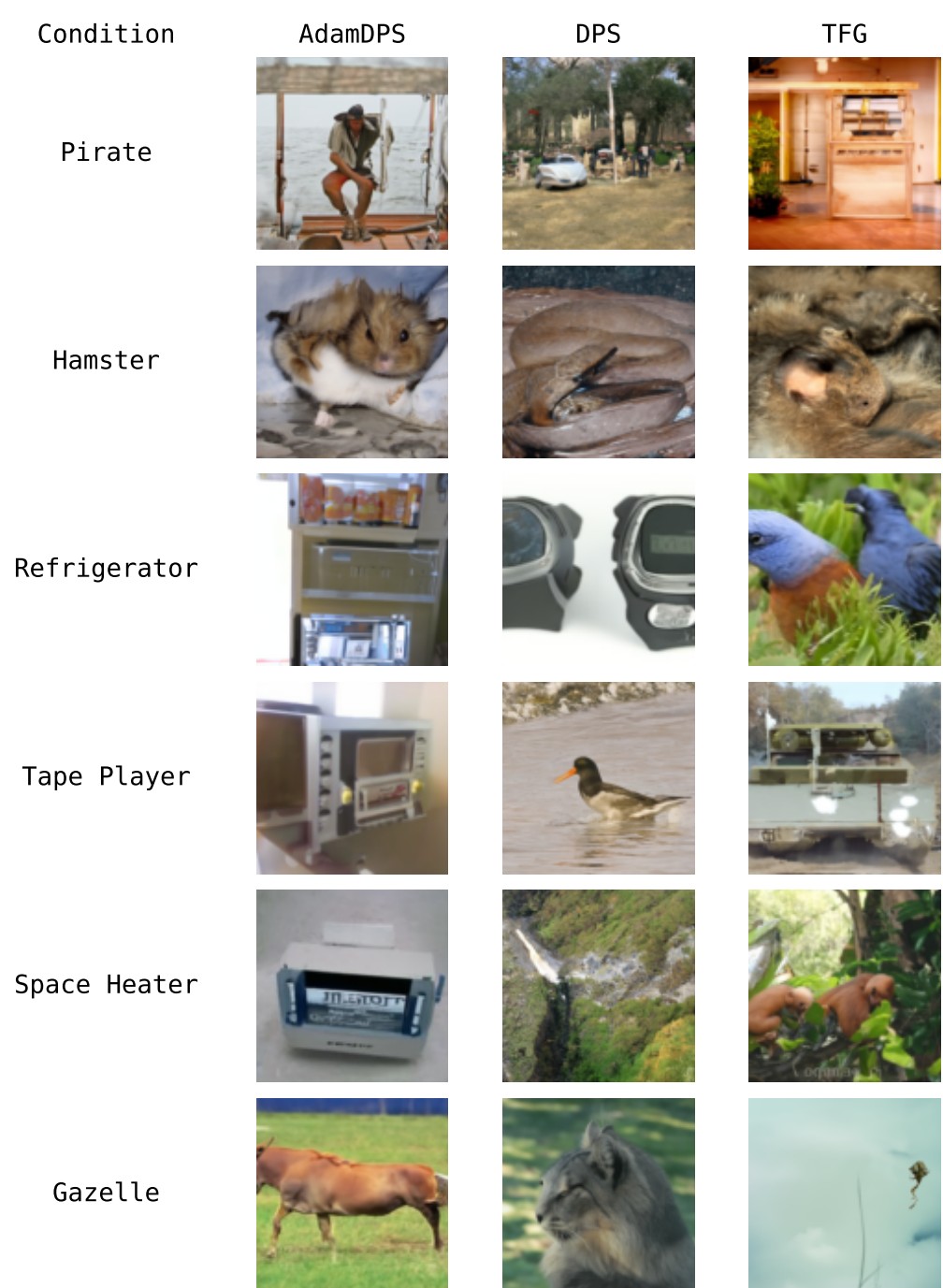

Figure 17: Additional qualitative results for class-conditional sampling with a standard classifier on ImageNet.

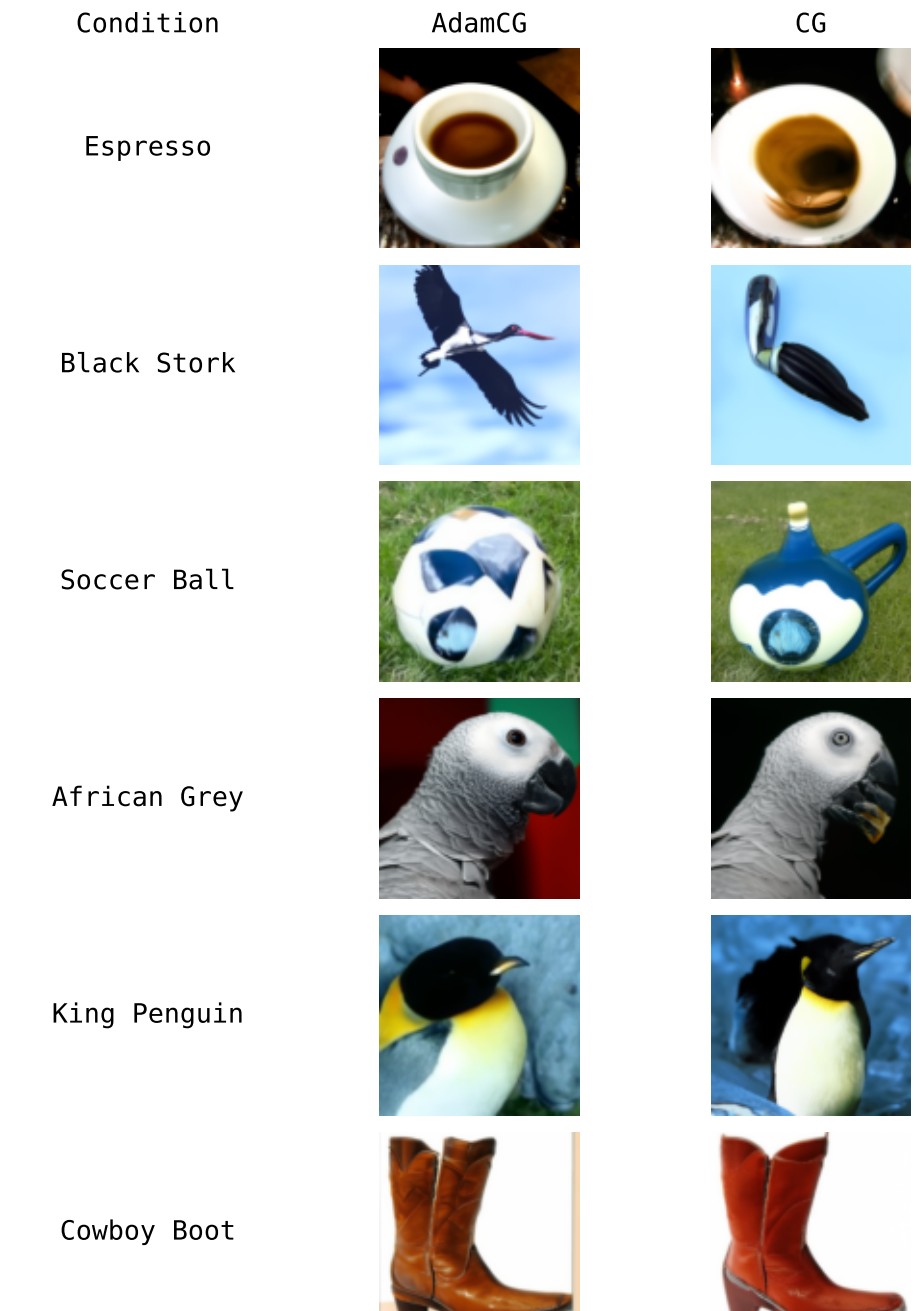

Figure 18: Additional qualitative results for class-conditional sampling with a time-aware classifier on ImageNet.

