# OpenReview forum: "Adaptive Moments are Surprisingly Effective for Plug-and-Play Diffusion Sampling"
_ICLR.cc/2026/Conference — ICLR 2026 Poster_

### Official Review · Reviewer_FmMn · 2025-10-29

**Soundness:** 2
**Presentation:** 1
**Contribution:** 1
**Rating:** 2
**Confidence:** 5

**Summary:**

This paper proposes a very simple plug-in modification to plug-and-play diffusion guidance: keep exponential moving averages of the likelihood-gradient during sampling (Adam-style first/second moments) and use the stabilized gradient to guide updates. The authors instantiate this on DPS ("AdamDPS") and on classifier guidance ("AdamCG"), provide a 2-D GMM toy study, and report empirical gains on various inverse problems.

**Strengths:**

- The change is minimal (Adam-style moments around the guidance gradient) and easy to graft onto existing DPS/CG code paths.

- The toy GMM illustrates how DPS becomes unstable under noisy likelihood gradients whereas AdamDPS stabilizes trajectories; several restoration benchmarks also show improvements in LPIPS/FID vs. DPS and some training-free baselines.

**Weaknesses:**

- Technical novelty is minimal. The paper just adds Adam-style gradient update to DPS/CG at inference time. There is no new objective, solver, or estimator; the contribution is only an optimizer wrapper around an existing guidance gradient (Algorithms 1–2). This is a straightforward, well-known idea from stochastic optimization brought over without new theory specific to diffusion guidance beyond intuition. This falls short of ICLR's bar for conceptual advance. The empirical results are useful but do not compensate for the lack of technical depth or new insight into diffusion guidance.

- Potential overfitting in Fig. 5. The combination of highest accuracy with worst FID suggests the guidance may be overfitting to the evaluation classifier rather than improving image quality. Classification accuracy alone does not imply good samples if the images are classifier-friendly but perceptually poor. Please use a different (held-out) classifier for evaluation than the one used during guidance/training, and report cross-model accuracy (and, ideally, human or ImageReward/VQA scores) to confirm the gains are not due to classifier overfitting.

**Questions:**

- Can you replicate gains on other plug-and-play frameworks (e.g., DDRM, PiGDM, MPGD-style data-space updates) and other backbones/samplers to support the "surprisingly effective" claim beyond DPS/CG?

- Beyond demonstrating that AdamDPS smooths noisy guidance, is there substantive novelty beyond applying an optimizer wrapper to DPS/CG?

---

> ### Author Response · Authors · 2025-11-22
>
> (W1) Using Adam to correct noisy plug-and-play guidance is a novel technical contribution. Prior work has focused on more complex approximations, while we show mitigating noise in existing methods drastically improves guidance (setting a new state of the art). This work introduces a fundamentally different direction for improving guidance methods.
>
> It is often the case that the components of a method are not new and it is the combination that advances the field. Notable examples of this,
>  * Residual connections existed before ResNet.
>  * Attention mechanisms existed before Transformers.
>  * Momentum and RMSprop (exponentially decaying average of squared gradients) both existed before Adam.
>
> This argument even extends to many of the existing plug-and-play guidance algorithms.
>  * LGD (ICML 2023 Poster) is DPS + Monte Carlo.
>  * MPGD (ICLR 2024 Poster) is dataspace gradient descent.
>  * UGD (ICLR 2024 poster) is DPS + MPGD + FreeDoM.
>  * TFG (Neurips 2024 Spotlight) is LGD + DPS + MPGD + UGD + FreeDoM.
>
> As is the case with all the above examples, in this paper we show that the right combination of existing components produces a novel contribution to the field.
>
> (W2) There may be a misunderstanding of the objective of class-conditional sampling, which is not to “improve image quality” but to draw samples from the data distribution that align with the target class. We show we are the only method that achieves nontrivial performance on ImageNet with a standard classifier. Further we provide qualitative results in Figures [14, 15, 16], which show the images generated are not adversarial images but reasonable looking generations that are consistent with the target label.
>
> During rebuttal we benchmarked a number of held-out classifiers from HuggingFace and found the results to be consistent with the results in our submission. Demonstrating that “overfitting” is not occurring to any meaningful degree.
>
> Held-out Classifier Results for CIFAR10 (Standard Classifier)
>
> | Method   |   ViT Finetuned Accuracy |   ViT Accuracy |   ConvNeXt Finetuned Accuracy |
> |:---------|-------------------------:|---------------:|------------------------------:|
> | TFG (1)  |                    0.388 |          0.375 |                         0.394 |
> | TFG (2)  |                    0.147 |          0.154 |                         0.170  |
> | TFG (4)  |                    0.167 |          0.171 |                         0.194 |
> | DPS      |                    0.403 |          0.394 |                         0.428 |
> | AdamDPS  |                    0.512 |          0.532 |                         0.526 |
>
> Held-out Classifier Results for ImageNet (Standard Classifier)
>
> | Method   |   DINO v2 Giant Top-10 Accuracy |   ViT Base Top-10 Accuracy |   MobileViT v2 Top-10 Accuracy |
> |:---------|--------------------------------:|---------------------------:|-------------------------------:|
> | TFG (1)  |                           0.009 |                      0.009 |                          0.011 |
> | TFG (2)  |                           0.006 |                      0.008 |                          0.011 |
> | TFG (4)  |                           0.009 |                      0.008 |                          0.010  |
> | DPS      |                           0.012 |                      0.008 |                          0.010  |
> | AdamDPS  |                           0.083 |                      0.092 |                          0.076 |
>
> Held-out Classifier Results for ImageNet (Time-Aware Classifier)
>
> | Method   |   DINO v2 Giant Accuracy |   ViT Base Accuracy |   MobileViT v2 Accuracy |
> |:---------|-------------------------:|--------------------:|------------------------:|
> | CG       |                     0.58 |                0.63 |                    0.51 |
> | AdamCG   |                     0.78 |                0.84 |                    0.68 |
>
> CIFAR10 Held-out Classifier HuggingFace Names
> * aaraki/vit-base-patch16-224-in21k-finetuned-cifar10
> * nateraw/vit-base-patch16-224-cifar10
> * ahsanjavid/convnext-tiny-finetuned-cifar10
>
> ImageNet Held-out Classifier HuggingFace Names
> * facebook/dinov2-giant-imagenet1k-1-layer
> * anonymous-429/osf-vit-base-patch16-224-imagenet
> * apple/mobilevitv2-1.0-imagenet1k-256

---

> ### Author Response · Authors · 2025-11-22
>
> (Q1) Yes. We see the greatest performance gains for AdamPiGDM. AdamMPGD does not have as consistent performance improvements, which is expected as MPGD is fundamentally limited since it only does dataspace updates which are highly sensitive to the quality of the condition $y$.
>
> AdamMPGD & AdamPiGDM Ablation on ImageNet Easy and Hard Reconstruction Tasks
>
> | Task                         | Method    | FID   | LPIPS |
> |:-----------------------------|:----------|------:|------:|
> | Super Resolution 4x          | TFG       | 25.37 | 0.141 |
> | Super Resolution 4x          | TFG       | 25.03 | 0.131 |
> | Super Resolution 4x          | TFG       | 26.62 | 0.140 |
> | Super Resolution 4x          | MPGD      | 27.75 | 0.155 |
> | Super Resolution 4x          | AdamMPGD  | 27.06 | 0.149 |
> | Super Resolution 4x          | PiGDM     | 63.73 | 0.480 |
> | Super Resolution 4x          | AdamPiGDM | 22.42 | 0.140 |
> | Super Resolution 4x          | DPS       | 26.43 | 0.193 |
> | Super Resolution 4x          | AdamDPS   | 20.92 | 0.116 |
> |------------------------------|-----------|-------|-------|
> | Super Resolution 16x         | TFG       | 42.27 | 0.322 |
> | Super Resolution 16x         | TFG       | 42.69 | 0.322 |
> | Super Resolution 16x         | TFG       | 42.92 | 0.317 |
> | Super Resolution 16x         | MPGD      | 52.66 | 0.350 |
> | Super Resolution 16x         | AdamMPGD  | 56.51 | 0.350 |
> | Super Resolution 16x         | PiGDM     | 49.41 | 0.484 |
> | Super Resolution 16x         | AdamPiGDM | 33.75 | 0.284 |
> | Super Resolution 16x         | DPS       | 35.55 | 0.295 |
> | Super Resolution 16x         | AdamDPS   | 30.16 | 0.269 |
> |------------------------------|-----------|-------|-------|
> | Gaussian Deblur Intensity 3  | TFG       | 27.21 | 0.202 |
> | Gaussian Deblur Intensity 3  | TFG       | 26.01 | 0.174 |
> | Gaussian Deblur Intensity 3  | TFG       | 25.74 | 0.162 |
> | Gaussian Deblur Intensity 3  | MPGD      | 31.18 | 0.225 |
> | Gaussian Deblur Intensity 3  | AdamMPGD  | 29.96 | 0.202 |
> | Gaussian Deblur Intensity 3  | PiGDM     | 65.04 | 0.484 |
> | Gaussian Deblur Intensity 3  | AdamPiGDM | 27.30 | 0.196 |
> | Gaussian Deblur Intensity 3  | DPS       | 27.66 | 0.229 |
> | Gaussian Deblur Intensity 3  | AdamDPS   | 23.45 | 0.166 |
> |------------------------------|-----------|-------|-------|
> | Gaussian Deblur Intensity 12 | TFG       | 47.34 | 0.363 |
> | Gaussian Deblur Intensity 12 | TFG       | 46.44 | 0.355 |
> | Gaussian Deblur Intensity 12 | TFG       | 49.55 | 0.363 |
> | Gaussian Deblur Intensity 12 | MPGD      | 52.11 | 0.382 |
> | Gaussian Deblur Intensity 12 | AdamMPGD  | 51.43 | 0.370 |
> | Gaussian Deblur Intensity 12 | PiGDM     | 51.97 | 0.504 |
> | Gaussian Deblur Intensity 12 | AdamPiGDM | 35.91 | 0.300 |
> | Gaussian Deblur Intensity 12 | DPS       | 40.25 | 0.341 |
> | Gaussian Deblur Intensity 12 | AdamDPS   | 38.35 | 0.328 |
>
> We use the implementation of PiGDM in https://github.com/devzhk/InverseBench, and tune the hyperparameter (i.e. guidance strength) with Bayesian optimization - the same way we tune all methods in our submission.

---

> ### Author Response · Authors · 2025-11-22
>
> (Q1 Continued) We also ran additional sampling step ablations with DDPM on ImageNet and with DDIM on ImageNet and Cats.
>
> DDPM ImageNet
>
> | Method  | Inference Steps | FID    | LPIPS |
> |:--------|----------------:|-------:|------:|
> | TFG (4) |              12 | 118.59 | 0.497 |
> | DPS     |              12 | 154.29 | 0.576 |
> | AdamDPS |              12 |  98.71 | 0.464 |
> |---------|-----------------|--------|-------|
> | TFG (4) |              25 |  52.35 | 0.389 |
> | DPS     |              25 |  62.21 | 0.424 |
> | AdamDPS |              25 |  49.05 | 0.347 |
> |---------|-----------------|--------|-------|
> | TFG (4) |              50 |  44.55 | 0.342 |
> | DPS     |              50 |  44.54 | 0.331 |
> | AdamDPS |              50 |  35.63 | 0.297 |
> |---------|-----------------|--------|-------|
> | TFG (4) |             100 |  42.92 | 0.317 |
> | DPS     |             100 |  35.55 | 0.295 |
> | AdamDPS |             100 |  30.16 | 0.269 |
>
> DDIM ImageNet
>
> | Method  | Inference Steps | FID   | LPIPS |
> |:--------|----------------:|------:|------:|
> | TFG (4) |              12 | 60.34 | 0.408 |
> | DPS     |              12 | 58.01 | 0.573 |
> | AdamDPS |              12 | 77.33 | 0.445 |
> |---------|-----------------|-------|-------|
> | TFG (4) |              25 | 53.84 | 0.352 |
> | DPS     |              25 | 51.34 | 0.451 |
> | AdamDPS |              25 | 46.35 | 0.348 |
> |---------|-----------------|-------|-------|
> | TFG (4) |              50 | 59.20 | 0.352 |
> | DPS     |              50 | 43.62 | 0.337 |
> | AdamDPS |              50 | 37.18 | 0.304 |
> |---------|-----------------|-------|-------|
> | TFG (4) |             100 | 58.38 | 0.349 |
> | DPS     |             100 | 41.50 | 0.314 |
> | AdamDPS |             100 | 39.60 | 0.293 |
>
> DDIM Cats
>
> | Method  | Inference Steps | FID   | LPIPS |
> |:--------|----------------:|------:|------:|
> | TFG (4) |              12 | 47.33 | 0.336 |
> | DPS     |              12 | 67.90 | 0.507 |
> | AdamDPS |              12 | 61.72 | 0.365 |
> |---------|-----------------|-------|-------|
> | TFG (4) |              25 | 49.85 | 0.333 |
> | DPS     |              25 | 42.73 | 0.367 |
> | AdamDPS |              25 | 32.51 | 0.294 |
> |---------|-----------------|-------|-------|
> | TFG (4) |              50 | 51.70 | 0.340 |
> | DPS     |              50 | 32.35 | 0.291 |
> | AdamDPS |              50 | 27.20 | 0.271 |
> |---------|-----------------|-------|-------|
> | TFG (4) |             100 | 55.81 | 0.354 |
> | DPS     |             100 | 30.31 | 0.292 |
> | AdamDPS |             100 | 27.53 | 0.267 |

---

> ### Author Response · Authors · 2025-11-22
>
> (Q2) We offer a number of important new observations in the paper.
>  * The typical benchmarks for plug-and-play diffusion sampling are very easy, making them misleading benchmarks for the development of new sampling methods.
>  * [1][2][3][4] all show results beating DPS, however, if the task is made slightly more challenging DPS beats all of these methods.
>  * Adaptive moments estimates can mitigate the impact of noisy guidance achieving state of the art performance across a range of datasets and tasks.
>
> The direction of the literature for plug-and-play guidance has long been to improve the likelihood score approximation itself. This led to much more complex approximations. We show there is an alternative direction for improving plug-and-play guidance by reducing the noise present in the guidance term.
>
> [1] Song, J., Zhang, Q., Yin, H., Mardani, M., Liu, M. Y., Kautz, J., ... & Vahdat, A. (2023, July). Loss-guided diffusion models for plug-and-play controllable generation. In International Conference on Machine Learning (pp. 32483-32498). PMLR.
>
> [2] He, Y., Murata, N., Lai, C. H., Takida, Y., Uesaka, T., Kim, D., ... & Ermon, S. (2024). Manifold Preserving Guided Diffusion. In The Twelfth International Conference on Learning Representations.
>
> [3] Ye, H., Lin, H., Han, J., Xu, M., Liu, S., Liang, Y., ... & Ermon, S. (2024). Tfg: Unified training-free guidance for diffusion models. Advances in Neural Information Processing Systems, 37, 22370-22417.
>
> [4] Mardani, M., Song, J., Kautz, J., & Vahdat, A. (2023). A variational perspective on solving inverse problems with diffusion models. arXiv preprint arXiv:2305.04391.

---

> ### Comment · Reviewer_FmMn · 2025-11-26
>
> Thank you for the detailed rebuttal and additional experiments.
>
> On novelty, I appreciate the clarification that your goal is to open a new “noise reduction” direction for plug-and-play guidance. However, my concern was never simply that the method is a combination of existing components. Many impactful works (ResNets, Transformers, Adam, and prior plug-and-play methods like LGD/MPGD/UGD/TFG) also combine known ideas, but they typically either introduce a new estimator, a substantially new modeling or variational view, or a clearly non-obvious architectural principle. In contrast, the core change here remains: take an existing guidance gradient and wrap it with Adam-style EMA at inference time. This feels like a relatively straightforward transfer of a well-known optimization heuristic, and the paper still does not provide a comparably strong conceptual or theoretical advance specific to diffusion guidance.
>
> Your new results are convincing. The held-out classifier evaluations largely address my overfitting concern in the class-conditional setting, and the additional ablations with AdamPiGDM/AdamMPGD and different samplers/backbones strengthen the empirical case that the idea generalizes beyond DPS/CG. However, the substantial FID degradation in class-conditional sampling is still concerning: even if the formal objective is class alignment, a method whose samples are well-classified but systematically of lower perceptual quality has limited practical value, and a more explicit analysis of this alignment–fidelity trade-off would be useful.
>
> Overall, I view the empirical contribution more favorably after the rebuttal, but I do not see the technical advance as sufficient for acceptance. I have modestly adjusted my scores to reflect this, while keeping my overall recommendation as rejection.

---

> > ### Author Response · Authors · 2025-12-03
> >
> > This response from FmMn is concerning as it contains factual errors. Notably, both the original review and this response from FmMn are classified as “Fully AI-Generated” by Pangram and GPT Zero.
> >
> > The reviewer states: "Many impactful works (ResNets, Transformers, Adam, and prior plug-and-play methods like LGD/MPGD/UGD/TFG) also combine known ideas, but they typically either introduce a new estimator, a substantially new modeling or variational view..."
> >
> > This claim is demonstrably incorrect. These examples were taken directly from our rebuttal where we specifically curated them as methods that do not introduce new estimators or modeling perspectives, but rather combine existing components.
> >
> > * ResNets = ConvNets + Residual Connections (both pre-existing)
> > * Transformers = MLPs + Attention Mechanisms (both pre-existing)
> > * Adam = Momentum + RMSprop (both pre-existing)
> > * LGD = DPS + Monte Carlo (both pre-existing)
> > * MPGD = Dataspace gradient descent (pre-existing)
> > * UGD = DPS + MPGD + FreeDoM (all pre-existing)
> > * TFG = LGD + DPS + MPGD + UGD + FreeDoM (all pre-existing)
> >
> > The reviewer has mischaracterized these examples to support an argument they actually refute. This is concerning as it suggests a fundamental misunderstanding of the literature.
> >
> > Regarding class-conditional sampling. The reviewer suggests our "FID degradation" limits practical value. This suggests a misunderstanding of the results. For class-conditional sampling with a standard classifier on ImageNet all baselines perform at random, while ours is the only method that achieves nontrivial performance. That is, AdamDPS is the only method that can perform this task. The baseline methods have no practical value as they completely fail to align with the condition. Further the qualitative results in Figures 14-16 show realistic, class-aligned images suggesting the drop in FID is not an issue with the “perceptual quality” as the reviewer suggests.
> >
> > Finally, the reviewer continues to overlook what is possibly the most important contribution of our work. We find that the typical benchmarks for plug-and-play diffusion sampling are misleadingly easy. Many of the recent advances in this area of research use these easy tasks as validation. We find all methods we benchmarked that were published after DPS, fail to beat DPS in harder settings. This is an important re-evaluation of existing work which has important implications for future work in this area — in particular it highlights a need for more rigorous evaluation.
> >
> > The problems with FmMn’s response which we discuss above demonstrate a failure to meaningfully engage with our work.

---

### Official Review · Reviewer_WUrY · 2025-11-01

**Soundness:** 2
**Presentation:** 3
**Contribution:** 2
**Rating:** 2
**Confidence:** 4

**Summary:**

The paper investigates the classifier guidance and finds out that classifier guidance approximates the likelihood score which often includes a lot of noise. The paper proposes to regularize the guidance sampling process with ADAM, a popular method in neural network optimization.

**Strengths:**

1. The paper is well written
2. the method is easy to understand

**Weaknesses:**

1. The novelty might be the concern. The Adam is not novel and the guidance is also not novel. The use of Adam during sampling process is also not new (this paper has similar idea https://www.ijcai.org/proceedings/2024/0157.pdf). The only difference is that Adam is applied to guidance term instead of denoising term as in the paper.
2. The paper does not provide a new fundamental observations or scientific hypothesis.
3. In terms of applications, the proposed method currently could not extend to classifier-free guidance which is a very popular guidance right now.
4. The quantitative results are not provided, different metrics are not considered e.g FID, IS, Rec, Prec, CLIP, GenEval, T2I Bench.

**Questions:**

Please see the weaknesses.

---

> ### Author Response · Authors · 2025-11-22
>
> (W1) We will add [1] to our related work. As the reviewer points out, this work does the opposite of what we propose. [1] wraps the denoising estimate in an adaptive moment estimate. We found that the denoising estimate is already very good as the diffusion model is explicitly trained for this, and that adding adaptive moments to the denoising estimate does not help.
>
> We will emphasize in our final draft that many of these ideas existed before: loss-guided sampling, Adam, DPS, etc. However, using Adam to correct noisy plug-and-play guidance is new.
>
> Further it is often the case that the components of a method are not new and it is the combination that advances the field. Notable examples of this,
>  * Residual connections existed before ResNet.
>  * Attention mechanisms existed before Transformers.
>  * Momentum and RMSprop (exponential moving average of squared gradients) both existed before Adam.
>
> This argument even extends to many of the existing plug-and-play guidance algorithms.
>  * LGD (ICML 2023 Poster) is DPS + Monte Carlo.
>  * MPGD (ICLR 2024 Poster) is dataspace gradient descent.
>  * UGD (ICLR 2024 poster) is DPS + MPGD + FreeDoM.
>  * TFG (Neurips 2024 Spotlight) is LGD + DPS + MPGD + UGD + FreeDoM.
>
> As is the case with the above examples, in this paper we show that the right combination of existing components produces a novel contribution to the field.
>
> (W2) We offer a number of important new observations in the paper.
>  * The typical benchmarks for plug-and-play diffusion sampling are very easy, making them misleading benchmarks for the development of new sampling methods.
>  * [2][3][4][5][8] all show results beating DPS, however, if the task is made slightly more challenging DPS beats all of these methods.
>  * Adaptive moments estimates can mitigate the impact of noisy guidance achieving state of the art performance across a range of datasets and tasks.
>
> The direction of the literature for plug-and-play guidance has long been to improve the likelihood score approximation itself. This led to much more complex approximations. We show there is an alternative direction for improving plug-and-play guidance by reducing the noise present in the guidance term.
>
> (W3) There may be a misunderstanding of the literature. Classifier-free guidance is indeed very popular, but it is only applicable in a dramatically different setting. Classifier-free guidance requires training a conditional diffusion model, which requires having a large amount of both computational resources and labeled data. Plug-and-play guidance is widely applicable as it requires no training, and no labeled data - only a guidance model. It remains an extremely important problem. [3] (ICLR 2024 Poster), [8] (ICLR 2024 Poster), [4] (Neurips 2024 Spotlight), [7] (ICLR 2025 Poster), [6] (ICLR 2025 Spotlight), are just a few works on plug-and-play guidance published in the past two years.
>
> [1] Wang, X., Dinh, A. D., Liu, D., & Xu, C. (2024, August). Boosting Diffusion Models with an Adaptive Momentum Sampler. In IJCAI (pp. 1416-1424).
>
> [2] Song, J., Zhang, Q., Yin, H., Mardani, M., Liu, M. Y., Kautz, J., ... & Vahdat, A. (2023, July). Loss-guided diffusion models for plug-and-play controllable generation. In International Conference on Machine Learning (pp. 32483-32498). PMLR.
>
> [3] He, Y., Murata, N., Lai, C. H., Takida, Y., Uesaka, T., Kim, D., ... & Ermon, S. (2024). Manifold Preserving Guided Diffusion. In The Twelfth International Conference on Learning Representations.
>
> [4] Ye, H., Lin, H., Han, J., Xu, M., Liu, S., Liang, Y., ... & Ermon, S. (2024). Tfg: Unified training-free guidance for diffusion models. Advances in Neural Information Processing Systems, 37, 22370-22417.
>
> [5] Mardani, M., Song, J., Kautz, J., & Vahdat, A. (2023). A variational perspective on solving inverse problems with diffusion models. arXiv preprint arXiv:2305.04391.
>
> [6] Zheng, H., Chu, W., Zhang, B., Wu, Z., Wang, A., Feng, B. T., ... & Yue, Y. (2025). Inversebench: Benchmarking plug-and-play diffusion priors for inverse problems in physical sciences. arXiv preprint arXiv:2503.11043.
>
> [7] Rissanen, S., Heinonen, M., & Solin, A. (2025). Free Hunch: Denoiser Covariance Estimation for Diffusion Models Without Extra Costs. In The Thirteenth International Conference on Learning Representations.
>
> [8] Bansal, A., Chu, H. M., Schwarzschild, A., Sengupta, R., Goldblum, M., Geiping, J., & Goldstein, T. (2024). Universal Guidance for Diffusion Models. In The Twelfth International Conference on Learning Representations.

---

> ### Author Response · Authors · 2025-11-22
>
> (W4) This is clearly an oversight by the reviewer. We already provide extensive quantitative results in Figures [1-3, 5-8] and Tables [1-7]. We provide FID numbers in all settings. We do not benchmark any text-to-image tasks so CLIP Score, GenEval, and T2I Bench are not applicable. We computed IS during rebuttal, and include it here.
>
> ImageNet Difficulty Ablation (with Inception Score)
>
> | Task                 | Method  | IS     | FID   | LPIPS |
> |:---------------------|:--------|-------:|------:|------:|
> | Super Resolution 4x  | TFG (1) | 132.58 | 25.37 | 0.141 |
> | Super Resolution 4x  | TFG (2) | 136.73 | 25.03 | 0.131 |
> | Super Resolution 4x  | TFG (4) | 128.90 | 26.62 | 0.140 |
> | Super Resolution 4x  | DPS     | 128.03 | 26.43 | 0.193 |
> | Super Resolution 4x  | AdamDPS | 180.91 | 20.92 | 0.116 |
> |----------------------|---------|--------|-------|-------|
> | Super Resolution 8x  | TFG (1) |  79.87 | 31.76 | 0.224 |
> | Super Resolution 8x  | TFG (2) |  80.09 | 31.56 | 0.219 |
> | Super Resolution 8x  | TFG (4) |  76.58 | 32.26 | 0.216 |
> | Super Resolution 8x  | DPS     |  88.64 | 29.02 | 0.234 |
> | Super Resolution 8x  | AdamDPS | 117.35 | 24.36 | 0.187 |
> |----------------------|---------|--------|-------|-------|
> | Super Resolution 12x | TFG (1) |  52.95 | 41.54 | 0.270 |
> | Super Resolution 12x | TFG (2) |  57.60 | 37.08 | 0.272 |
> | Super Resolution 12x | TFG (4) |  53.69 | 39.84 | 0.271 |
> | Super Resolution 12x | DPS     |  71.02 | 32.63 | 0.257 |
> | Super Resolution 12x | AdamDPS |  85.87 | 27.60 | 0.226 |
> |----------------------|---------|--------|-------|-------|
> | Super Resolution 16x | TFG (1) |  46.69 | 42.27 | 0.322 |
> | Super Resolution 16x | TFG (2) |  46.63 | 42.69 | 0.322 |
> | Super Resolution 16x | TFG (4) |  45.20 | 42.92 | 0.317 |
> | Super Resolution 16x | DPS     |  60.02 | 35.55 | 0.295 |
> | Super Resolution 16x | AdamDPS |  65.69 | 30.16 | 0.269 |
>
> | Task                         | Method  | IS     | FID   | LPIPS |
> |:-----------------------------|:--------|-------:|------:|------:|
> | Gaussian Deblur Intensity 3  | TFG (1) | 113.49 | 27.21 | 0.202 |
> | Gaussian Deblur Intensity 3  | TFG (2) | 125.65 | 26.01 | 0.174 |
> | Gaussian Deblur Intensity 3  | TFG (4) | 128.00 | 25.74 | 0.162 |
> | Gaussian Deblur Intensity 3  | DPS     | 109.29 | 27.66 | 0.229 |
> | Gaussian Deblur Intensity 3  | AdamDPS | 144.23 | 23.45 | 0.166 |
> |------------------------------|---------|--------|-------|-------|
> | Gaussian Deblur Intensity 6  | TFG (1) |  63.57 | 35.43 | 0.285 |
> | Gaussian Deblur Intensity 6  | TFG (2) |  72.71 | 33.21 | 0.262 |
> | Gaussian Deblur Intensity 6  | TFG (4) |  73.29 | 33.58 | 0.260 |
> | Gaussian Deblur Intensity 6  | DPS     |  71.41 | 31.39 | 0.262 |
> | Gaussian Deblur Intensity 6  | AdamDPS |  90.08 | 27.26 | 0.228 |
> |------------------------------|---------|--------|-------|-------|
> | Gaussian Deblur Intensity 9  | TFG (1) |  46.36 | 42.91 | 0.331 |
> | Gaussian Deblur Intensity 9  | TFG (2) |  49.85 | 40.15 | 0.318 |
> | Gaussian Deblur Intensity 9  | TFG (4) |  49.63 | 39.58 | 0.308 |
> | Gaussian Deblur Intensity 9  | DPS     |  54.54 | 36.65 | 0.304 |
> | Gaussian Deblur Intensity 9  | AdamDPS |  62.14 | 33.63 | 0.280 |
> |------------------------------|---------|--------|-------|-------|
> | Gaussian Deblur Intensity 12 | TFG (1) |  36.11 | 47.34 | 0.363 |
> | Gaussian Deblur Intensity 12 | TFG (2) |  38.92 | 46.44 | 0.355 |
> | Gaussian Deblur Intensity 12 | TFG (4) |  38.74 | 49.55 | 0.363 |
> | Gaussian Deblur Intensity 12 | DPS     |  45.07 | 40.25 | 0.341 |
> | Gaussian Deblur Intensity 12 | AdamDPS |  49.11 | 38.35 | 0.328 |

---

### Official Review · Reviewer_sSbi · 2025-11-03

**Soundness:** 2
**Presentation:** 3
**Contribution:** 2
**Rating:** 4
**Confidence:** 4

**Summary:**

This paper studies the likelihood approximation noise in training-free plug-and-play diffusion models such as DPS. These models estimate the posterior score using $\nabla_xt log⁡p(xt∣y)=\nabla_xt log⁡p(y∣xt) + \nabla_xt log⁡p(xt)$, where the likelihood term $\nabla_{x_t}\log p(y|x_t)$ is approximated by a differentiable surrogate $\nabla_{x_t}L(f_\phi(\hat x_{0|t}),y)$. At high noise levels (early and mid diffusion steps), this estimate becomes highly unstable, leading to biased and noisy sampling.

The paper proposes maintaining exponential moving averages (EMAs) of the first and second moments of the likelihood gradient, similar to Adam. By tracking running averages of the gradient and its squared values across timesteps and adaptively rescaling updates, the approach stabilizes the guidance direction and reduces variance. Two variants are presented: AdamDPS for diffusion posterior sampling and AdamCG for classifier guidance.

Experiments demonstrate consistent improvements: on inpainting, deblurring, and $16\times$ super-resolution tasks (ImageNet and Cat datasets), AdamDPS achieves 1–2 dB PSNR gains and lower LPIPS/FID than DPS, UGD, and LGD. On CIFAR-10 and ImageNet 64×64 class-conditional generation, AdamCG improves top-1 and top-10 accuracy without significant degradation in FID. Synthetic 2-D Gaussian mixture experiments confirm that adaptive moments smooth the gradient variance peak occurring at intermediate diffusion steps.

**Strengths:**

Addresses a key limitation—noisy and biased likelihood gradients—with a simple, general, and low-cost modification.

Empirically robust across datasets, with clear ablations on noise scales and $(\beta_1, \beta_2)$.

Minimal implementation effort with measurable gains in quality and stability.

**Weaknesses:**

Limited to gradient-based plug-and-play samplers; not applicable to variational or optimization-based methods such as RED-Diff.

No theoretical analysis showing that adaptive moments preserve unbiasedness or convergence to the correct posterior $p(x_0)\exp[-r(x_0,y)]$.

Missing comparisons with RED-Diff, MPGD, and TMPD, which already incorporate momentum or adaptive scaling.

No examination of how the moving averages interact with the diffusion dynamics or affect the stationary distribution.


RED-Diff — M. Mardani, J. Song, J. Kautz, A. Vahdat. A Variational Perspective on Solving Inverse Problems with Diffusion Models, arXiv:2305.04391 (2023).

MPGD — Y. He, N. Murata, C.-H. Lai, Y. Takida, T. Uesaka, D. Kim, W.-H. Liao, Y. Mitsufuji, J. Z. Kolter, R. Salakhutdinov, S. Ermon. Manifold Preserving Guided Diffusion, ICLR 2024.

TMPD — B. Boys, et al. Tweedie Moment Projected Diffusions for Inverse Problems, arXiv:2310.06721 (2023).

**Questions:**

How does AdamDPS fundamentally differ from optimization-based samplers such as RED-Diff that already employ Adam updates?

Does maintaining EMAs across timesteps bias the effective sampling trajectory?

How would AdamDPS perform against RED-Diff or TMPD under the same compute and step budgets?

Could combining adaptive moments with Monte-Carlo smoothing (e.g., LGD) provide further gains?

How sensitive is the performance to the number of diffusion steps or when applied to ODE-based samplers such as DDIM?

---

> ### Author Response · Authors · 2025-11-22
>
> (W1) Our method and RED-diff belong to different categories of approaches to plug-and-play sampling. MPGD, TFG, FreeDoM, UGD, etc. are all not applicable to optimization-based methods like RED-diff. In the same way that ResNet is not applicable to DenseNet, they are just two different things and that is not a weakness.
>
> Our method is compatible with many guidance techniques. We show this is the case for DPS, and CG in the paper. During rebuttals we show this for MPGD, and PiGDM. It could potentially be combined with many different guidance methods: LGD, FreeDoM, UGD, TFG, etc.
>
> (W2) We agree that theoretical analysis of Adam in the context of diffusion sampling would be valuable and will add discussion of this to the paper as future work. The primary, and significant, contributions of this work are extensive empirical analysis of both Adam for guidance and the typical guidance benchmarks.
>  * We show AdamDPS sets the state-of-the-art in numerous settings, please see Figures [3, 5] and Tables [1-7].
>  * We show the performance of existing methods (e.g. TFG, MPGD, UGD, etc.) degrades significantly as the task becomes more challenging, please see Figure [6] and the ImageNet difficulty ablation provided in our response to reviewer br5n.
>  * We show that DPS is more robust than existing methods, outperforming existing methods in challenging settings, please see Figure [3, 5, 6], Tables [1, 3, 5].
>  * We show AdamDPS makes DPS more robust to noisy guidance, ultimately improving the alignment of the stationary distribution with the ground truth, please see Figures [1, 2].
>  * We provide qualitative and quantitative analysis of the sampling trajectories, please see Figures [8, 9]
>  * We provide qualitative results in Figures [10-16]
>
> (W3 & Q3) We already provide comparisons to MPGD in our submission, please see Figure 3 and Tables [1, 3, 5, 6]. We have added the requested comparison to RED-diff.
>
> RED-diff Benchmarked on ImageNet Easy and Hard Reconstruction Tasks
>
> | Task                         | Method   | IS     | FID    | LPIPS |
> |:-----------------------------|:---------|-------:|-------:|------:|
> | Super Resolution 4x          | TFG (1)  | 132.58 |  25.37 | 0.141 |
> | Super Resolution 4x          | TFG (2)  | 136.73 |  25.03 | 0.131 |
> | Super Resolution 4x          | TFG (4)  | 128.90 |  26.62 | 0.140 |
> | Super Resolution 4x          | RED-diff | 102.09 |  30.42 | 0.201 |
> | Super Resolution 4x          | DPS      | 128.03 |  26.43 | 0.193 |
> | Super Resolution 4x          | AdamDPS  | 180.91 |  20.92 | 0.116 |
> |------------------------------|----------|--------|--------|-------|
> | Super Resolution 16x         | TFG (1)  |  46.69 |  42.27 | 0.322 |
> | Super Resolution 16x         | TFG (2)  |  46.63 |  42.69 | 0.322 |
> | Super Resolution 16x         | TFG (4)  |  45.20 |  42.92 | 0.317 |
> | Super Resolution 16x         | RED-diff |   9.66 | 100.30 | 0.569 |
> | Super Resolution 16x         | DPS      |  60.02 |  35.55 | 0.295 |
> | Super Resolution 16x         | AdamDPS  |  65.69 |  30.16 | 0.269 |
> |------------------------------|----------|--------|--------|-------|
> | Gaussian Deblur Intensity 3  | TFG (1)  | 113.49 |  27.21 | 0.202 |
> | Gaussian Deblur Intensity 3  | TFG (2)  | 125.65 |  26.01 | 0.174 |
> | Gaussian Deblur Intensity 3  | TFG (4)  | 128.00 |  25.74 | 0.162 |
> | Gaussian Deblur Intensity 3  | RED-diff |  76.44 |  40.13 | 0.271 |
> | Gaussian Deblur Intensity 3  | DPS      | 109.29 |  27.66 | 0.229 |
> | Gaussian Deblur Intensity 3  | AdamDPS  | 144.23 |  23.45 | 0.166 |
> |------------------------------|----------|--------|--------|-------|
> | Gaussian Deblur Intensity 12 | TFG (1)  |  36.11 |  47.34 | 0.363 |
> | Gaussian Deblur Intensity 12 | TFG (2)  |  38.92 |  46.44 | 0.355 |
> | Gaussian Deblur Intensity 12 | TFG (4)  |  38.74 |  49.55 | 0.363 |
> | Gaussian Deblur Intensity 12 | RED-diff |   5.85 | 209.90 | 0.525 |
> | Gaussian Deblur Intensity 12 | DPS      |  45.07 |  40.25 | 0.341 |
> | Gaussian Deblur Intensity 12 | AdamDPS  |  49.11 |  38.35 | 0.328 |
>
> We use the implementation of RED-diff in https://github.com/devzhk/InverseBench, and tune the hyperparameters (i.e. learning rate, $\beta_1$, $\beta_2$, and $\lambda$) with Bayesian optimization - the same way we tune all methods in our submission.

---

> ### Author Response · Authors · 2025-11-22
>
> (W4) We provide a number of empirical analyses on the effect of adaptive moments throughout the paper.
>  * We show the effect of AdamDPS on the stationary distribution in our synthetic study. Please see Figure 1.
>  * We measure FID in all settings which measures the difference between the sampled distribution of images and the ground truth distribution of images. In nearly all settings AdamDPS has the best FID. Please see Figures [3, 5], Tables [1-7], and the ImageNet difficulty ablation provided in our response to reviewer br5n.
>  * We show the guidance loss throughout sampling to help characterize the sampling dynamics of the different methods. Please see Figure 8.
>  * We also provide qualitative analysis via the clean data predictions throughout sampling to help characterize the sampling dynamics of the different methods. Please see Figure 9.
>
> (Q1) There may be a misunderstanding of the literature, since AdamDPS and RED-diff are quite different approaches. AdamDPS is a modification that can be applied to any of the ODE or SDE solvers typically used for sampling. An example is sampling with annealed Langevin Dynamics, where the diffusion timesteps are run in reverse and at each step noise is removed according to both the diffusion model and the guidance loss before a smaller amount of noise is added back — AdamDPS only applies adaptive moments to the guidance loss.
>
> RED-diff solves an optimization problem instead of performing a sampling procedure. The objective they minimize is the guidance loss with a weighted regularization term encouraging alignment with the diffusion model estimates. RED-diff solves this optimization problem with Adam, so the gradient of the entire objective is wrapped in the adaptive moment estimate. This means the denoising step sizes are no longer determined by the diffusion process but by the optimizer.
>
> (Q2) In most settings we see no evidence of this. The exception is for class-conditional sampling with a standard classifier, please see our discussion with reviewer br5n about this.
>
> (Q4) It’s absolutely possible to add adaptive moments to LGD. We did not try this as DPS alone consistently outperforms LGD in difficult settings.
>
> (Q5) We already provided a sampling step ablation of the 16x super resolution task on the Cats dataset with DDPM sampling. Please see Figure 7 (left subplot). During rebuttal we ran the same sampling step ablation with DDPM on ImageNet and with DDIM sampling on both ImageNet and Cats.
>
> DDPM ImageNet
>
> | Method  | Inference Steps | FID    | LPIPS |
> |:--------|----------------:|-------:|------:|
> | TFG (4) |              12 | 118.59 | 0.497 |
> | DPS     |              12 | 154.29 | 0.576 |
> | AdamDPS |              12 |  98.71 | 0.464 |
> |---------|-----------------|--------|-------|
> | TFG (4) |              25 |  52.35 | 0.389 |
> | DPS     |              25 |  62.21 | 0.424 |
> | AdamDPS |              25 |  49.05 | 0.347 |
> |---------|-----------------|--------|-------|
> | TFG (4) |              50 |  44.55 | 0.342 |
> | DPS     |              50 |  44.54 | 0.331 |
> | AdamDPS |              50 |  35.63 | 0.297 |
> |---------|-----------------|--------|-------|
> | TFG (4) |             100 |  42.92 | 0.317 |
> | DPS     |             100 |  35.55 | 0.295 |
> | AdamDPS |             100 |  30.16 | 0.269 |
>
> DDIM ImageNet
>
> | Method  | Inference Steps | FID   | LPIPS |
> |:--------|----------------:|------:|------:|
> | TFG (4) |              12 | 60.34 | 0.408 |
> | DPS     |              12 | 58.01 | 0.573 |
> | AdamDPS |              12 | 77.33 | 0.445 |
> |---------|-----------------|-------|-------|
> | TFG (4) |              25 | 53.84 | 0.352 |
> | DPS     |              25 | 51.34 | 0.451 |
> | AdamDPS |              25 | 46.35 | 0.348 |
> |---------|-----------------|-------|-------|
> | TFG (4) |              50 | 59.20 | 0.352 |
> | DPS     |              50 | 43.62 | 0.337 |
> | AdamDPS |              50 | 37.18 | 0.304 |
> |---------|-----------------|-------|-------|
> | TFG (4) |             100 | 58.38 | 0.349 |
> | DPS     |             100 | 41.50 | 0.314 |
> | AdamDPS |             100 | 39.60 | 0.293 |
>
> DDIM Cats
>
> | Method  | Inference Steps | FID   | LPIPS |
> |:--------|----------------:|------:|------:|
> | TFG (4) |              12 | 47.33 | 0.336 |
> | DPS     |              12 | 67.90 | 0.507 |
> | AdamDPS |              12 | 61.72 | 0.365 |
> |---------|-----------------|-------|-------|
> | TFG (4) |              25 | 49.85 | 0.333 |
> | DPS     |              25 | 42.73 | 0.367 |
> | AdamDPS |              25 | 32.51 | 0.294 |
> |---------|-----------------|-------|-------|
> | TFG (4) |              50 | 51.70 | 0.340 |
> | DPS     |              50 | 32.35 | 0.291 |
> | AdamDPS |              50 | 27.20 | 0.271 |
> |---------|-----------------|-------|-------|
> | TFG (4) |             100 | 55.81 | 0.354 |
> | DPS     |             100 | 30.31 | 0.292 |
> | AdamDPS |             100 | 27.53 | 0.267 |

---

### Official Review · Reviewer_br5n · 2025-11-08

**Soundness:** 4
**Presentation:** 4
**Contribution:** 4
**Rating:** 8
**Confidence:** 3

**Summary:**

This paper introduces AdamDPS, a new plug-and-play method for diffusion sampling. By noting that the guided sampling update is gradient ascent using the score of the likelihood distribution, the authors introduce Adam-style moment exponential moving averages to stabilize the overall update. The resulting method is tested on a toy GMM example, demonstrating improvements in sampling quality over the baseline diffusion posterior sampling (DPS). Experiments are then extended to reconstruction tasks on image datasets, where AdamDPS demonstrates the best combination of both FID and LPIPs. Ablations are performed on task-difficulty, Adam hyperparams, and wall-clock time.

**Strengths:**

This paper is complete and well-presented. Overall, the method is effective and experiments are thorough. I would recommend it for acceptance.

**(S1)**: Simple and effective. The core idea of introducing Adam-style moment stabilization is simple and effective for diffusion guided plug-and-play sampling. The paper clearly motivates the problem with previous approaches, proposes a sound improvement, and demonstrates the improvement via experimental results.

**(S2)**: Clear improvement over prior work on reconstruction tasks. Results in Fig 3 clearly demonstrate superior reconstruction quality and sample quality of AdamDPS.

**(S3)**: Comprehensive ablations and analysis. The ablations on task difficulty is valuable and demonstrates the core motivation for the method-- as the task gets noisier, the stabilization introduced by Adam-style moments results in better performance. Other ablations on number of sampling steps and moment coefficient values are useful for any future users of this method.

**(S4)**: Computational efficiency. The ablation on wall-clock time confirms that this method induces little to no overhead over baselines, which is valuable.

**Weaknesses:**

**(W1)**: Slightly mixed results on class-conditional sampling. The FID on ImageNet and CIFAR-10 seems worse than baselines, even as the classification accuracy is much better.

**(W2)**: Slight regressions over baselines on easy tasks. In Fig 6, for easier super-resolution or deblurring tasks, AdamDPS is slightly worse than TFG. This again slightly clouds the otherwise clear narrative of the paper. No explanation for this is given.

**(W3)**: Details on the method are missing. A clearer explanation of the models used, experimental setup, metrics would be very valuable for clarity. For example, while an ablation on $\beta_1$ and $\beta_2$ were performed, it's not clear what the optimal values are for each task.

**Questions:**

**(Q1)**: Is there a missing increment to $k$ in algorithms 1 and 2?

---

> ### Author Response · Authors · 2025-11-22
>
> Thank you for your thoughtful engagement with our work.
>
> (W1) It’s true we do find there is an alignment vs. diversity tradeoff for class-conditional sampling with a standard classifier. Though FID is lower, the images still look realistic (see qualitative results in Figures [14, 15]). We suspect it has to do with the clean data estimates early in sampling all looking similar and nothing like the real image distribution a standard classifier is trained on, which results in similar but meaningless guidance. Alternatively it might be because we tune for CMMD with batch size 32, possibly a different objective or larger batch would help.
>
> (W2) Yes, for very easy tasks methods that perform dataspace update have an advantage. This is because the dataspace update is gradient descent on the predicted image with the condition $y$. When the condition $y$ is very informative, which is the case in easy settings, it is not surprising this works extremely well. We find that this advantage is lost when the task is made only slightly harder.
> The Cats dataset is generally quite easy for methods to do well on, so we have run the full difficulty ablation on ImageNet. We can see that AdamDPS outperforms nearly all other methods on all tasks in FID and LPIPS. The single exception is in the easiest deblur task where $N_{\text{iter}}=4$ wins by 0.004 in LPIPs but still loses in FID.
>
> ImageNet Difficulty Ablation
>
> | Task                 | Method  | FID   | LPIPS |
> |:---------------------|:--------|------:|------:|
> | Super Resolution 4x  | TFG (1) | 25.37 | 0.141 |
> | Super Resolution 4x  | TFG (2) | 25.03 | 0.131 |
> | Super Resolution 4x  | TFG (4) | 26.62 | 0.140 |
> | Super Resolution 4x  | DPS     | 26.43 | 0.193 |
> | Super Resolution 4x  | AdamDPS | 20.92 | 0.116 |
> |----------------------|---------|-------|-------|
> | Super Resolution 8x  | TFG (1) | 31.76 | 0.224 |
> | Super Resolution 8x  | TFG (2) | 31.56 | 0.219 |
> | Super Resolution 8x  | TFG (4) | 32.26 | 0.216 |
> | Super Resolution 8x  | DPS     | 29.02 | 0.234 |
> | Super Resolution 8x  | AdamDPS | 24.36 | 0.187 |
> |----------------------|---------|-------|-------|
> | Super Resolution 12x | TFG (1) | 41.54 | 0.270 |
> | Super Resolution 12x | TFG (2) | 37.08 | 0.272 |
> | Super Resolution 12x | TFG (4) | 39.84 | 0.271 |
> | Super Resolution 12x | DPS     | 32.63 | 0.257 |
> | Super Resolution 12x | AdamDPS | 27.60 | 0.226 |
> |----------------------|---------|-------|-------|
> | Super Resolution 16x | TFG (1) | 42.27 | 0.322 |
> | Super Resolution 16x | TFG (2) | 42.69 | 0.322 |
> | Super Resolution 16x | TFG (4) | 42.92 | 0.317 |
> | Super Resolution 16x | DPS     | 35.55 | 0.295 |
> | Super Resolution 16x | AdamDPS | 30.16 | 0.269 |
>
> | Task                         | Method  | FID   | LPIPS |
> |:-----------------------------|:--------|------:|------:|
> | Gaussian Deblur Intensity 3  | TFG (1) | 27.21 | 0.202 |
> | Gaussian Deblur Intensity 3  | TFG (2) | 26.01 | 0.174 |
> | Gaussian Deblur Intensity 3  | TFG (4) | 25.74 | 0.162 |
> | Gaussian Deblur Intensity 3  | DPS     | 27.66 | 0.229 |
> | Gaussian Deblur Intensity 3  | AdamDPS | 23.45 | 0.166 |
> |------------------------------|---------|-------|-------|
> | Gaussian Deblur Intensity 6  | TFG (1) | 35.43 | 0.285 |
> | Gaussian Deblur Intensity 6  | TFG (2) | 33.21 | 0.262 |
> | Gaussian Deblur Intensity 6  | TFG (4) | 33.58 | 0.260 |
> | Gaussian Deblur Intensity 6  | DPS     | 31.39 | 0.262 |
> | Gaussian Deblur Intensity 6  | AdamDPS | 27.26 | 0.228 |
> |------------------------------|---------|-------|-------|
> | Gaussian Deblur Intensity 9  | TFG (1) | 42.91 | 0.331 |
> | Gaussian Deblur Intensity 9  | TFG (2) | 40.15 | 0.318 |
> | Gaussian Deblur Intensity 9  | TFG (4) | 39.58 | 0.308 |
> | Gaussian Deblur Intensity 9  | DPS     | 36.65 | 0.304 |
> | Gaussian Deblur Intensity 9  | AdamDPS | 33.63 | 0.280 |
> |------------------------------|---------|-------|-------|
> | Gaussian Deblur Intensity 12 | TFG (1) | 47.34 | 0.363 |
> | Gaussian Deblur Intensity 12 | TFG (2) | 46.44 | 0.355 |
> | Gaussian Deblur Intensity 12 | TFG (4) | 49.55 | 0.363 |
> | Gaussian Deblur Intensity 12 | DPS     | 40.25 | 0.341 |
> | Gaussian Deblur Intensity 12 | AdamDPS | 38.35 | 0.328 |
>
> (W3) Thank you for pointing this out. We will include in the camera ready version of the paper a section that provides the full experimental details. We will also release the complete code to reproduce our results. All models and data used are publicly available.
>
> (Q1) The variable k is updated in the AdaptiveMomentEstimate function call, but we forgot to include the pseudocode for the AdaptiveMomentEstimate function in the submission. We will make sure to include all relevant pseudocode in the camera ready version of the paper.

---

> > ### Comment · Reviewer_br5n · 2025-11-27
> >
> > Thanks! I'm already in favor of acceptance, so I will maintain my score.

---

### Author Response · Authors · 2025-12-03

**Summary**

**Problematic Reviews**
* Reviews and official comments from reviewers WUrY, sSbi, and FmMn contain concerning factual errors and major oversights. These are primarily discussed in our private AC comment, and our response to FmMn’s comment on our rebuttal. Notably, all interactions with sSbi, and FmMn are classified as “Fully AI-Generated” by Pangram and GPT Zero. Despite marking high confidence the reviewers repeatedly demonstrate a failure to engage with our work.

**Two Important Contributions**
* **Benchmarking Reality Check**: We find that the typical benchmarks for plug-and-play diffusion sampling are misleadingly easy. Many of the recent advances in this area of research use these easy tasks as validation. We benchmark these recent works in more challenging settings and find their performance falls below one of the simplest baselines, DPS. This has important implications for future work on plug-and-play diffusion sampling.
* **Simple yet SOTA**: We find adaptive moments reduce the noise in the guidance term, which drastically improves a range of existing guidance methods (e.g. DPS, CG, PiGDM). These Adam augmented guidance algorithms set new state-of-the-art results on a variety of tasks and datasets, outperforming a collection of recent—more complicated—guidance algorithms.

**Strengths**
* Simple to understand and implement (br5n, sSbi, WUrY, FmMn).
* Consistent improvements over prior work on reconstruction tasks (br5n, FmMn).
* Method is broadly applicable (sSbi, FmMn).
* Computationally efficient (br5n, sSbi).
* Comprehensive ablations (br5n, sSbi).
* Comprehensive analysis (br5n).
* Paper is well written (WUrY).

**Weaknesses (Addressed during Rebuttal)**
* For the easiest reconstruction tasks on the Cats dataset AdamDPS falls below TFG-2 and TFG-4. We found that for very easy settings, methods that perform dataspace updates have an advantage. We added results for the same tasks but on ImageNet, which show AdamDPS beats TFG-2 in all metrics on all tasks, and only loses to TFG-4 by 0.004 LPIPS in the easiest Gaussian Deblurring settings—winning in all others. Please see our response to br5n.
* We added a comparison to RED-diff in a variety of settings as requested. These results strengthen the case for AdamDPS, as it remains the dominant method. Please see response to sSbi.
* We added additional sampling step ablations. The submission already has a sampling step ablation for DDPM on Cats. We added DDIM on Cats, as well as DDPM and DDIM on ImageNet. Please see response to sSbi.
* We added Inception Score results as requested. These results provide further evidence in support of AdamDPS. Please see our response to WUrY.
* We added evaluation of the class-conditional results with heldout classifiers as requested. The results provide further evidence in support of AdamDPS, and AdamCG. Please see our response to FmMn.
* We added a comparison to PiGDM, as well as benchmarked AdamMPGD and AdamPiGDM. These results show that adaptive moments can drastically improve existing guidance algorithms, setting new state-of-the-art on a variety of tasks and datasets. Please see our response to FmMn.

**Weaknesses (Partially Addressed during Rebuttal)**
* For class-conditional sampling with a standard classifier AdamDPS achieves markedly higher accuracy though there is some degradation in FID. Specifically on ImageNet, AdamDPS is the only method that achieves class alignment better than random chance. That is, all prior works completely fail to align with the condition. Further the qualitative results in Figures 14-16 show realistic, class-aligned images. We provide some discussion of why FID might degrade in our response to reviewer br5n.

---

### Meta-Review · Area_Chair_jRmG · 2026-01-06

**Summary:**

The manuscript proposed using adaptive moment estimation to stablize noisy likelihood scores in guidance models. The paper is largely empirical and demonstrates that the method, despite being quite simple, achieves good performance.

**Reviewer Concerns:**

The authors addressed most concerns by the reviewers.

**Reviewer Scores:**

With the extensive new numerical tests and comparisons, I suspect that the reviewers might increase the scores.

---

### Decision · Program_Chairs · 2026-01-26

Accept (Poster)